# Mycorrhizal C/N ratio determines plant-derived carbon and nitrogen allocation to symbiosis

Rodica Pena[1,2], Sarah L. Bluhm[3], Silke Ammerschubert[1], Paola Agüi-Gonzalez [4], Silvio O. Rizzoli [4], Stefan Scheu[3,5] & Andrea Polle [1,5✉]

Carbon allocation of trees to ectomycorrhizas is thought to shape forest nutrient cycling, but the sink activities of different fungal taxa for host resources are unknown. Here, we investigate fungal taxon-specific differences in naturally composed ectomycorrhizal (EM) communities for plant-derived carbon and nitrogen. After aboveground dual labeling of young beech with $^{15}$N and $^{13}$C, ectomycorrhizas formed with different fungal taxa exhibit strong differences in label enrichment. Secondary Ion Mass Spectrometry (SIMS) imaging of nitrogen in cross sections of ectomycorrhizas demonstrates plant-derived $^{15}$N in both root and fungal structures. Isotope enrichment in ectomycorrhizas correlates with that in the corresponding ectomycorrhiza-attached lateral root, supporting fungal taxon-specific N and C fluxes in ectomycorrhizas. The enrichments with $^{13}$C and $^{15}$N in the symbiosis decrease with increasing C/N ratio of ectomycorrhizas, converging to zero at high C/N. The relative abundances of EM fungal species on roots are positively correlated with $^{13}$C enrichment, demonstrating higher fitness of stronger than of less C-demanding symbioses. Overall, our results support that differences among the C/N ratios in ectomycorrhizas formed with different fungal species regulate the supply of the symbioses with host-derived carbon and provide insights on functional traits of ectomycorrhizas, which are important for major ecosystem processes.

[1] Forest Botany and Tree Physiology, University of Göttingen, Göttingen, Germany. [2] Department of Sustainable Land Management, School of Agriculture Policy and Development, University of Reading, Reading, UK. [3] J.F. Blumenbach Institute of Zoology and Anthropology, Animal Ecology, University of Göttingen, Göttingen, Germany. [4] Department of Neuro- and Sensory Physiology and Center for Biostructural Imaging of Neurodegeneration, University Medical Center Göttingen, Göttingen, Germany. [5] Centre for Biodiversity and Sustainable Land Use, University of Göttingen, Göttingen, Germany. ✉email: apolle@gwdg.de

In temperate forest ecosystems, roots of dominant tree species are colonized by diverse assemblages of ectomycorrhizal (EM) fungal species[1–5]. EM fungi form mutualistic associations with root tips, called ectomycorrhizas. Ectomycorrhizas act as an interface for nutrient fluxes between soil and trees[6,7], where fungi transfer low-availability nutrients from the soil to the host trees and receive carbohydrates in return[8–11]. Thus, ectomycorrhizas link the belowground carbon (C) allocation and nutrient dynamics in forest ecosystems[12–14]. Although their role is paramount in soil C accumulation[7,15], fungal traits that drive host resource allocation to different fungal taxa within composed assemblages remain poorly understood[16]. It has been long acknowledged that the supply of host carbohydrates is required to maintain the extensive diversity of EM fungal assemblages[17,18] that commonly occurs in individual trees[19,20]. In contrast to other plant mutualistic associations, such as arbuscular mycorrhizae[21,22] or legume–rhizobia[23], where the plant carbohydrate supply is correlated with the magnitude of fungal or bacterial nutrient delivery, a direct reciprocal recompense has not been observed in ectomycorrhizas[24–27].

In vascular plants, the transport of photoassimilates from the source to sink organs primarily depends on sink demand. Ectomycorrhizas are new organs formed by fungal and root cells at the root tips[28]. In ectomycorrhizas, the plant-derived carbohydrate supply provides the substrate for the maintenance of both the plant and fungal cells, including the energy-consuming processes of nutrient uptake, transformation, and transfer. The fungus may induce an additional C sink in the ectomycorrhizas as compared with non-EM root tips[29,30], but this is not always the case[26], leaving an open question about the size of and contributing factors to C sink in ectomycorrhizas.

After uptake and translocation from roots to leaves, inorganic nitrogen (N) is used for amino acid biosynthesis and is then redistributed within the plant from source to sink tissues. The primary source of N used for root growth is the phloem-transported organic N from the leaves[31]. Whether ectomycorrhizas formed with different fungal taxa have distinct effects on internal tree N allocation to root tips remains largely unknown.

The main objective of this study was to examine nutrient concentrations and to determine the sink activity for host resources of ectomycorrhizas. We investigated the incorporation of C and N originating from aboveground leaf assimilation into ectomycorrhizas formed with different fungal taxa in naturally composed assemblages. Since N and C metabolism are linked, and N accumulation varies profoundly among different EM fungal species[27,32–34], we reasoned that plant C transfer to an EM fungal species is determined not only by fungal biomass[26] but also by fungal taxon-specific N utilization. Hence, C provision to an ectomycorrhiza would be related to its taxon-specific C/N stoichiometry. We hypothesized that an EM fungal species with a high C and low N content would be unlikely to have a high demand for host C in contrast to an EM fungal species with low C and high N content. In contrast to C, we expected that host N allocation to ectomycorrhizas reflects the demand of the root cells in the symbiosis and, therefore, would not vary in ectomycorrhizas formed with different fungal species. However, given the large variation among EM species in N acquisition[34] and the unknown impact on the N content in the root cells in the symbiosis, we also considered the alternative hypothesis that tree-derived N transfer to ectomycorrhizas is affected by EM fungal species.

To address these hypotheses, we used beech (*Fagus sylvatica*), representing the most abundant deciduous tree species in Central European temperate forests[35]. We pulse-labeled young trees colonized with natural EM assemblages aboveground with $^{15}NH_4^+$ and $^{13}CO_2$ and linked the enrichment of the isotopes

with the C and N content in EM fungal species and the relative abundance of the EM fungal species on plant root tips.

## Results

**N and C are translocated from aboveground tissues into ectomycorrhizal root tips**. After dual labeling with $^{15}NH_4^+$ (selected leaves) and $^{13}CO_2$ (aboveground aerial space), each for 3 days, we found significant enrichment of both labels in all tissues studied along transport path in bark close to the stem bottom, in different parts of the root system, and in ectomycorrhizas (Fig. 1). No changes of $^{13}C$ enrichment occurred in different root parts between day 5 and day 20 (Fig. 1a), whereas $^{15}N$ enrichment slightly increased in day 20 in comparison with day 5 (Fig. 1b). The total amounts of newly acquired $^{13}C$ and $^{15}N$ in roots were $154.1 \pm 18.8$ mg $^{13}C$ and $142.2 \pm 29.2$ µg $^{15}N$ (means across both sampling dates of $n = 10$ plants $\pm$ SE). The share of $^{13}C$ and $^{15}N$ in fine roots in the pools were $27.1 \pm 4.4$ mg $^{13}C$ and $23.3 \pm 4.2$ µg $^{15}N$.

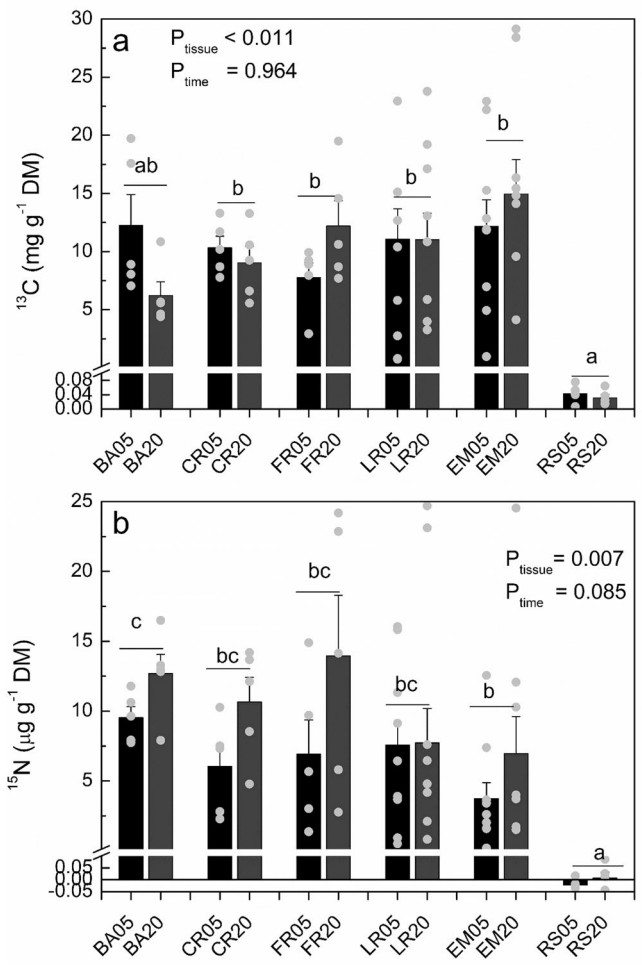

**Fig. 1 Enrichment of $^{13}C$ and $^{15}N$ in the plant tissues, ectomycorrhizas and rhizosphere soil. a** Enrichment of $^{13}C$, **b** enrichment of $^{15}N$ in bark (BA), coarse roots (CR), fine roots (FR), lateral rootlets (LR), ectomycorrhizas (EM), and rhizosphere soil (RS) 5 and 20 days after the start of $^{13}C$ pulse labeling. Symbols represent individual measurements. Bars show means ($n = 10$ per time point and tissue for lateral roots and ectomycorrhizal, $n = 5$ per time point and tissue for bark, coarse and fine roots, ±SE). Different letters above bars indicate significant differences ($P \leq 0.05$) among label concentrations in different compartments (GLM, HSD post hoc test).

The highest N concentrations were detected in ectomycorrhizas and the highest C concentrations in lateral rootlets (Table 1).

The least $^{13}$C enrichment occurred in the rhizosphere, exceeding, however, the level of non-labeled controls (Null hypothesis, $P = 0.002$) (Fig. 1a). The rhizosphere $^{13}$C amount was $1.9 \pm 0.3$ mg per root system (means across both sampling dates of $n = 10$ plants $\pm$SE). No significant enrichment in the rhizosphere was observed for $^{15}$N (Null hypothesis, $P = 0.457$) (Fig. 1b).

**Ectomycorrhizas exhibit strong fungal taxon-related differences in plant-derived C and N transfer.** The beech trees were colonized by 16 different EM fungal species (Fig. 2), whose assembly structures did not vary between day 5 and day 20 (ANOSIM, $P = 0.414$). Neither root tip vitality (mean: $89.6 \pm 2.1\%$, Student's t-test, $P = 0.772$) nor mycorrhizal colonization (mean: $92.6 \pm 2.2\%$, Student's t-test, $P = 0.718$) differed between the sampling dates. Among the 16 EM fungal taxa, seven were sufficiently abundant for $^{13}$C and $^{15}$N analyses and present at each sampling date. These species colonized together about 80% of the inspected EM root tips (Fig. 2). The estimated proportion of fungal tissue in EM root tips ranged from 25.3% (UEM3) to 50.0% (*Tomentella punicea*) with a mean of $34.0 \pm 3.5\%$ (Supplementary Table S1). The UEM (uncultured EM) fungi belonged to the order of Helotiales, which is mainly known for endophytes and saprotrophic species and less for mycorrhizas[36]. Therefore, we inspected cross sections and

confirmed the presence of a Hartig net and hyphal mantle for the UEM species (Supplementary Fig. S2b).

Ectomycorrhizal root tips showed no significant time-dependent changes of $^{13}$C or $^{15}$N between day 5 and day 20 (Supplementary Table S2), suggesting that net fluxes of the tracers had ceased.

The fungal species *Pachyphlodes conglomerata* colonized about 30% of the root tips (Fig. 2). *Pachyphlodes* (Pezizaceae) are truffle-like fungi that form ectomycorrhizas mainly with Fagaceae[37]. They showed the strongest $^{13}$C enrichment, indicating that ectomycorrhizas formed with this EM species exhibited the strongest sink for $^{13}$C at the level of individual root tip and within the community (Fig. 3a, Supplementary Fig. S3a). Ectomycorrhizas with UEM1 (Helotiales) and the basidiomycetes *Tomentella punicea* showed the second strongest enrichment of $^{13}$C, while ectomycorrhizas with the other studied fungal species showed only moderate $^{13}$C enrichments (Fig. 3a, Supplementary Fig. S3a). Carbon concentrations in ectomycorrhizas associated with different fungal taxa ranged from about 380 to 440 mg C g$^{-1}$ dry mass (Fig. 3b).

The highest $^{15}$N enrichment was found in ectomycorrhizas formed with UEM1 (Fig. 3c). Ectomycorrhizas formed with *P. conglomerata* assumed an intermediate position, while $^{15}$N enrichment was low in all other ectomycorrhizas studied here (Fig. 3c). NanoSIMS imaging, using the most abundant ectomycorrhiza with *P. conglomerata* as an example, showed that $^{15}$N enrichment was distributed among root and fungal cells in symbiosis and was higher than the natural abundance of $^{15}$N in those tissues (Supplementary Fig. S4). Nitrogen concentrations varied considerably in ectomycorrhizas with different fungal species, ranging from 20 to 40 mg N g$^{-1}$ dry mass (Fig. 3d).

We tested whether the concentrations of C, N, or the enrichments in $^{13}$C or $^{15}$N in the ectomycorrhizas with different fungal species were related to the proportion of fungal tissue in the symbioses (as obtained from digital images of the cross sections, Supplementary Table S1). No significant relationships were detected (linear relationships of the fraction of fungal tissue with $^{13}$C: R = 0.475, $P = 0.282$, with C: R = 0.580, $P = 0.172$, with $^{15}$N: R = 0.176, $P = 0.704$, with N: R = 0.226, $P = 0.624$).

**Table 1 Nitrogen (N) and carbon (C) concentrations in different plant compartments and the rhizosphere.**

| Compartment | Abbreviation | N (mg g$^{-1}$ dry weight) | C (mg g$^{-1}$ dry weight) |
|---|---|---|---|
| Bark | BA | $8.75 \pm 0.17$ | $448.1 \pm 4.2$ |
| Coarse roots | CR | $6.41 \pm 0.16$ | $452.7 \pm 2.1$ |
| Fine roots | FR | $12.34 \pm 0.63$ | $344.3 \pm 15.1$ |
| Lateral roots | LR | $15.69 \pm 0.43$ | $471.0 \pm 3.3$ |
| Ectomycorrhiza | EM | $26.23 \pm 0.98$ | $419.7 \pm 4.7$ |
| Rhizosphere soil | RS | $2.64 \pm 0.24$ | $36.6 \pm 3.8$ |
| $P_{compartment}$ | | <0.001 | <0.001 |
| $P_{time}$ | | 0.125 | 0.459 |

Data show means of $n = 10$, except for lateral roots and ectomycorrhiza $n = 20$ ($\pm$SE). Different letters in columns indicate significant differences at $P < 0.05$ (GLM, HSD, time as random factor).

**Lateral rootlets uncover contrasting sink strength for nitrogen and carbon in ectomycorrhizas.** Lateral roots are part of the fine root fraction and connect higher-order root branches with the EM root tips (Supplementary Fig. S2a). We reasoned that a correlation between a newly acquired isotope in a lateral root and its attached ectomycorrhiza indicates an influence of the EM fungal species on resource allocation because the own C and N demand of lateral roots was expected to be independent on that of the associated symbiont. We found a significant linear relationship between $^{13}$C enrichment in lateral roots and that of the associated taxon-specific ectomycorrhizas (Fig. 4a). Notably, the slope of 1.23 was significantly higher than 1.0 (comparison of regression lines, $P = 0.016$), indicating a significant enhancement of $^{13}$C in ectomycorrhizas compared with the connected lateral root.

$^{15}$N enrichment in lateral roots was also significantly correlated with $^{15}$N enrichment in the associated taxon-specific symbiont (Fig. 4b) but the slope of the regression line was only 0.53 and significantly lower than 1.0 (comparison of regression lines: $P < 0.001$).

Based on the abundance and biomass of the EM species and their enrichments with $^{13}$C and $^{15}$N, we estimated the EM pool sizes of the transferred $^{13}$C and $^{15}$N as $1.59 \pm 0.19$ mg $^{13}$C and $0.52 \pm 0.08$ µg $^{15}$N (means of $n = 20$ plants). The contribution of the individual EM species to the total EM fungal pool size is shown in Supplementary Fig. S3a.

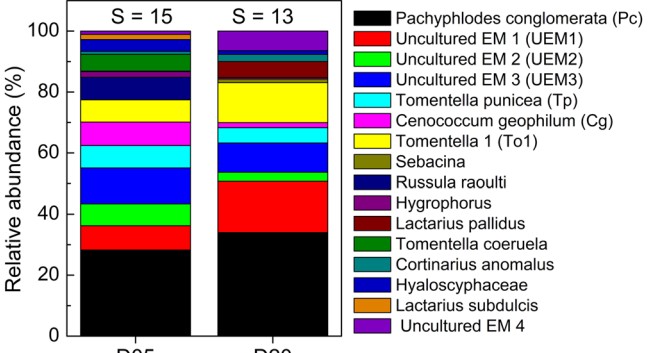

**Fig. 2 Relative abundance of ectomycorrhizal species colonizing the roots of young beech trees (*Fagus sylvatica*).** Measurements were conducted on 10 trees per sampling date (D05 and D20). S above the stacked bars indicates the total number of species detected per sampling date.

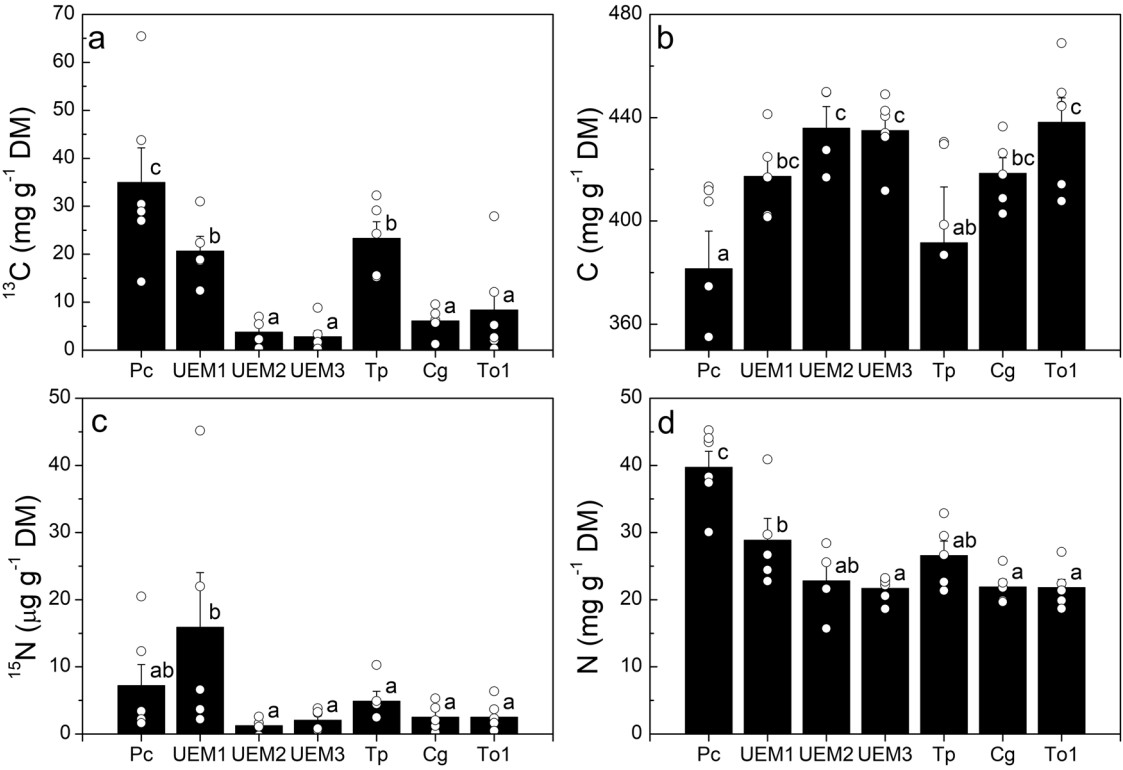

**Fig. 3 Enrichment of $^{13}$C and $^{15}$N and carbon and nitrogen concentrations in distinct ectomycorrhizal fungal species. a** $^{13}$C enrichment. **b** Carbon concentration. **c** $^{15}$N enrichment. **d** Nitrogen concentration. Ectomycorrhizal fungal species names are abbreviated as follows: Pc: *Pachyphlodes conglomerata* ($n = 6$), UEM1: uncultured EM 1 ($n = 5$), UEM2: uncultured EM 2 ($n = 5$), UEM3: uncultured EM 3 ($n = 6$), Tp: *Tomentella punicea* ($n = 5$), Cg: *Cenococcum geophilum* ($n = 7$), To1: *Tomentella* sp1 ($n = 6$). Symbols represent individual measurements. Number n in brackets indicate the number of individual trees from which the ectomycorrhizal fungal species were collected. Data are from day 5 and day 20. Bars show means (±SE). Different letters above bars indicate significant differences with $P \leq 0.05$ (ANOVA and post hoc test HSD).

**Ectomycorrhizal $^{13}$C and $^{15}$N allocation are correlated with C/N ratio.** We tested whether $^{13}$C or $^{15}$N enrichments in EM species were related to their C/N ratio. $^{13}$C enrichment decreased with increasing C concentrations (R = 0.741, $P < 0.001$, Supplementary Fig. S5a) and increased with increasing N concentrations in EM species (R = 0.833, $P < 0.001$, Supplementary Fig. S5b). As the result, $^{13}$C enrichment in ectomycorrhizas was negatively correlated with the C/N ratio of the symbiosis ($y = -2.49x + 57.1$, R = 0.832, $P = 0.001$, Fig. 5). We found a C/N threshold of about 24, above which almost no $^{13}$C allocation to EM species was observed (Fig. 5).

$^{15}$N enrichment showed a similar but weaker relationship with C/N ($y = -0.57x + 15.1$, R = 0.337, $P = 0.041$), which notably converged at the same C/N threshold (Fig. 5).

We further tested whether the amounts of $^{13}$C and $^{15}$N in the biomass of taxon-specific ectomycorrhizas were also related to their C/N ratios. We found a negative relationship of C/N with the EM taxon-specific amounts of $^{13}$C and a positive relationship with the EM taxon-specific amounts of $^{15}$N (Supplementary Fig. S3b, c).

**$^{13}$C enrichment in ectomycorrhizas is positively associated with ectomycorrhizal species abundance on roots.** We furthermore tested whether $^{13}$C allocation to ectomycorrhizas was related to the abundance of the respective symbiont. We found a positive relationship of $^{13}$C enrichment in EM species with their frequency of root fungal colonization (Fig. 6) but no relationship for $^{15}$N enrichment and root colonization (R = 0.233, $P = 0.165$)

## Discussion

In this study, we found a tight relationship between the enrichment of photo-assimilated C and C/N ratios of ectomycorrhizal species. This finding supports our hypothesis that EM nutrient balance is related to C allocation to colonized root tips. Furthermore, we found that EM fungal species receiving more C were colonizing more root tips than those receiving only little C. This result is ecologically important because the extent of root colonization is an indicator for the fitness of distinct EM fungal taxa.

We also show that a small but significant fraction of newly assimilated C appeared in the rhizosphere. A major fraction of the rhizosphere $^{13}$C was probably located in the extramatrical mycelia of ectomycorrhizas, in agreement with other studies[8,11,38–40]. This assumption is also supported by the finding that after aboveground $^{13}$C labeling, 30% of rhizosphere $^{13}$C was present in fungi, but only 1% in different prokaryotic organisms such bacteria and actinomycetes[41].

In our study, stronger $^{13}$C enrichment in ectomycorrhizas than in the lateral root, high EM colonization rates (>90%) and the occurrence of $^{13}$C in the rhizosphere corroborate that EM fungi are a sink for plant C[25,26]. We found that ~6% of the new $^{13}$C present in fine roots was located in the EM fraction. Assuming that total amount of $^{13}$C found in the rhizosphere also passed through the ectomycorrhizas, ~13% of the $^{13}$C in fine roots was allocated to ectomycorrhizas and their associated sinks. These fractions are expected to vary with the exposure time and are also likely affected by tree phenology because the direction of C flux depends on sink activities such as leaf and wood production in spring and summer, and nutrient resorption and allocation to

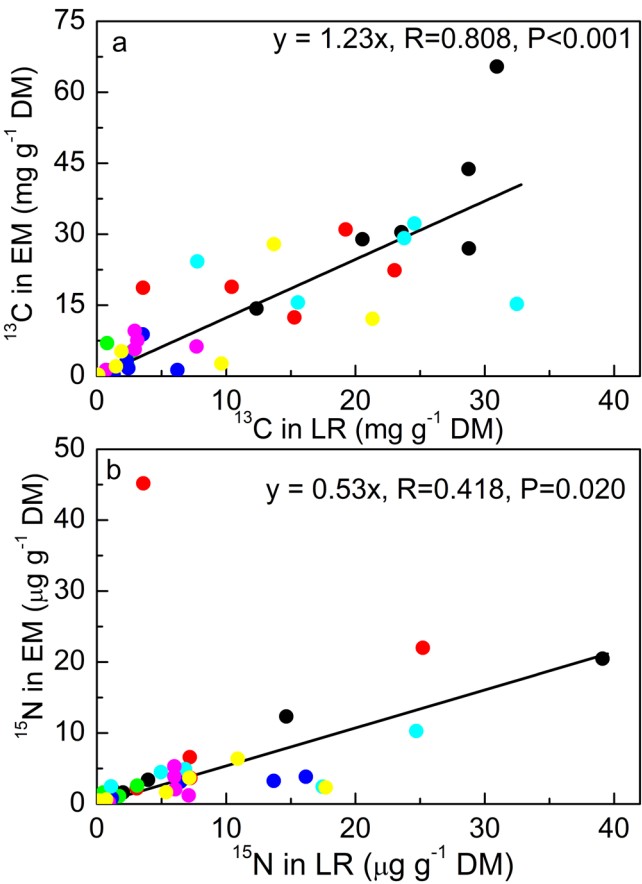

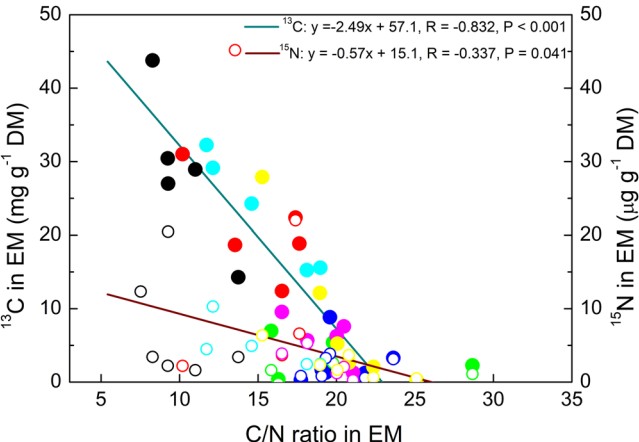

**Fig. 5 Relationship of $^{13}$C and $^{15}$N enrichment in ectomycorrhizas (EM) with their C/N ratio.** Circles refer to individual measurements of isotope enrichments in ectomycorrhizal fungal species per plant: *Pachyphlodes conglomerata* (black), UEM1: uncultured EM 1 (red), UEM2: uncultured EM 2 (green), UEM3: uncultured EM 3 (blue), Tp: *Tomentella punicea* (turquoise), Cg: *Cenococcum geophilum* (pink), To_1: Tomentella sp1 (yellow). Open symbols: $^{15}$N, closed symbols: $^{13}$C. Data are from day 5 and day 20.

**Fig. 4 Relationship of isotope enrichment in ectomycorrhizas with the isotope enrichment in the attached lateral rootlets. a** $^{13}$C. **b** $^{15}$N. Isotope data of a distinct ectomycorrhizal species were plotted against isotope data in the lateral rootlets (LR), which were connected with that ectomycorrhizal species. Symbols refer to individual measurements per plant. Colored symbols refer to the following ectomycorrhizal fungal species: *Pachyphlodes conglomerata* (black), UEM1: uncultured EM 1 (red), UEM2: uncultured EM 2 (green), UEM3: uncultured EM 3 (blue), Tp: *Tomentella punicea* (turquoise), Cg: *Cenococcum geophilum* (pink), To_1: *Tomentella* sp1 (yellow). Data are from day 5 and day 20.

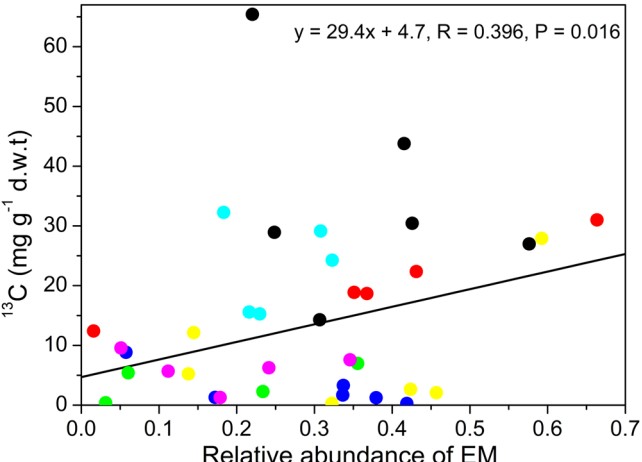

**Fig. 6 $^{13}$C enrichment in ectomycorrhizal fungal species in relation to their relative abundance on root tips.** Circles refer to individual measurements of $^{13}$C enrichments in ectomycorrhizal fungal species per plant. Symbol refer to: *Pachyphlodes conglomerata* (black), UEM1: uncultured EM 1 (red), UEM2: uncultured EM 2 (green), UEM3: uncultured EM 3 (blue), Tp: *Tomentella punicea* (turquoise), Cg: *Cenococcum geophilum* (pink), To_1: Tomentella sp1 (yellow). Data are from day 5 and day 20.

storage pools in fall[42,43]. In our study, which was conducted after the main phase of leaf and wood production, coarse roots were a great carbon sink, containing ~5 times more $^{13}$C than fine roots. Carbon stored in coarse roots can be used to supply ectomycorrhizas when recent photoassimilates are unavailable[18].

In the present experiment, nitrophilic fungi such as Pezizaceae, Helotiales, and Thelephoraceae[44] were the dominant species, most likely due to use of relatively N-rich soil (rhizosphere C/N = 13.6). Nevertheless, the taxon-specific ectomycorrhizas in these assemblages covered a wide C/N range from about 30 to 7, supporting that the specific fungal nutrient physiology rather than the N concentrations in soil determined fungal N and C concentrations. However, further experiments in soils covering a range of nutrient availabilities are necessary to examine this proposition.

Our findings have important implications for understanding plant-soil nutrient cycling. Högberg et al.[13] suggested a model for boreal forests, in which C allocation from trees to ectomycorrhizas drives the forest N cycle affecting soil N availability for plant nutrition by a feedback-loop. In nutrient-poor systems, the feedback-loop is driven towards increased N-limitations because high C allocation to the symbionts enables higher fungal N

immobilization. In contrast, under N-rich conditions, plants decrease C allocation to ectomycorrhizas, which triggers higher fungal N export to stabilize the association with the host[12,13,45]. Our study suggests limits for the operation of the proposed ectomycorrhiza-driven nutrient cycle because under our experimental conditions, the $^{13}$C allocation to ectomycorrhizas was almost zero when their C/N ratio was greater than 24. Our results imply that C allocation to ectomycorrhizas, which is not accompanied by N uptake and immobilization in the fungus, would lead to an increase in the EM C/N ratio and consequently result in reduced C flux to the symbiont. In line with this idea, ectomycorrhizas with *C. geophilum*, which exhibit lower N acquisition abilities than those with *P. conglomerata*, some Helotiales, or *T. punicea*[31–33], showed a low $^{13}$C enrichment and

a high C/N ratio in the present study. In general, the ability for N immobilization depends on EM species identity[32,33,46,47] and their N uptake systems[48–51]. Therefore, we speculate that fungal traits for N use efficiency may exert a feedback control on plant C allocation. Further investigation is required to examine the extent to which these fungal traits may vary during longer temporal trajectories and across different environments.

The result that plant-derived N was detected in fungal tissues in ectomycorrhizas deserves attention. Trees circulate N bidirectionally between roots and leaves in xylem and phloem, thereby enabling distribution of N at the whole-tree level between storage pools (mainly stem and coarse roots in deciduous trees) and active growth processes[52]. Increasing cellular N pools, especially of amino acids, suppress N transport systems and, thus, regulate tree N uptake[52]. Since N uptake occurs mainly close to the root apex[53], knowledge on allocation of host N to root tips is important. We found a correlation between [15]N enrichment and the C/N ratio of taxon-specific ectomycorrhizas. [15]N enrichment in the lateral rootlets corresponded to that in the attached EM fungal species, providing insights into independent regulation of N fluxes within the root system towards distinct sinks formed by different EM taxa. Since amino acids are unloaded from the phloem[54], with a proportion of them being exported to the root apoplast[55], it is possible that the observed [15]N label in ectomycorrhizas was caused by this mechanism. However, roots also produce and exude other N-bearing compounds apart from amino acids such as nucleosides, strigolactone, and alkaloids[56–59]. The biochemical identity of the transferred [15]N is, therefore, unclear.

An obvious question is how much N is allocated to ectomycorrhizas compared with C. Based on our data, we cannot provide a conclusive answer because of the time lapse between the [15]N and the [13]C labeling and the differences in label intensities. However, comparing the [13]C/[15]N ratios, which increased from 259 in roots to 3043 in ectomycorrhizas, in our experiment for each unit of [15]N the plant provided 11 units of [13]C to the EM root tips. Further studies are required to refine this analysis.

Notably, [15]N appeared also in fungal tissues of the ectomycorrhizas. To the best of our knowledge, our results represent the first evidence of plant-derived N presence in the fungal cells. In contrast to C, we found no relationship of plant N allocation to EM root tips and fungal colonization intensity. We can only speculate about the mechanisms resulting in plant-derived N in fungal tissues. EM fungi possess several amino acid transporters[60], implying that they have the potential to recapture the plant-exuded N. The absence of a significant [15]N enrichment in the rhizosphere may suggest that N was retained within plant-fungal symbiosis. This could indicate that ectomycorrhizas were more "tight" for N than for C or that [15]N was diluted by the fungal N pool, while passaging through extramatrical hyphae. Alternatively, it is plausible that plant-derived signaling or defense compounds were specifically targeted to fungal tissues in the ectomycorrhiza.

In conclusion, we show that the C/N ratios of EM fungal species were correlated with plant-derived C and N allocation to root tips (Fig. 7). The tight relationship between isotope enrichment in EM fungal-colonized root tips with corresponding lateral rootlets supports sink activity-driven fluxes of C and N to root tips. These sink activities were stronger for C than for N because the former showed net enrichment and the latter net depletion in EM fungal species compared with the attached rootlets. Overall, these results suggest that the ectomycorrhizal C/N stoichiometry may play a role in influencing the allocation of host resources to symbiosis. A further innovative result was the localization of plant-derived N in the fungal tissue of the symbiosis. Carbon but not N allocation to EM fungal species was related to host

colonization and was traced in the rhizosphere. These results fill a gap in our understanding of EM fungal-specific interactions with their host and enhance knowledge required to understand major ecosystem processes such as C and N cycling.

## Methods

**Collection of field-grown beech trees**. Young, healthy beech (*Fagus sylvatica*) trees ($n = 40$) with fully open leaves were excavated from May 10th to 15th in a deciduous forest (Billinghäuser Schlucht, Göttingen, Latitude: 51°35′15.39″, Longitude: 9°58′57.95″, 362 m above sea level, mean annual temperature 8.5 °C, mean sum of annual precipitation 614 mm). The trees were carefully removed and placed with intact soil layers individually in pots (diameter 183 mm, height 255 mm, corresponding to about 6.5 L soil volume). Soil has been classified as Orthic Rendzina on limestone with L-Mull as the humus type; the soil in this forest has a pH value of 4.9 and contains 36 mg C g$^{-1}$ and 2.6 mg N g$^{-1}$ dry weight soil[61]. Herbs were removed. The trees were kept for about 10 weeks under common garden conditions in Göttingen. The plants were watered regularly and were used in summer (end of July to August) for the labeling experiments. The trees were similar in size (±SE): 133 ± 4 cm height, 12.05 ± 0.29 mm width at the stem base, 0.23 ± 0.02 m$^2$ whole-plant leaf surface area, 7.3 ± 0.6 g leaf biomass, 12.5 ± 1.0 g coarse root and 3.9 ± 1.1 g fine root biomass.

**Stable isotope labeling of beech trees**. For [15]N leaf labeling, three adjacent top, three middle, and three bottom leaves were selected, whose surface was slightly abraded with fine grained abrasive paper (Basic Korn 240, LUX, Wermelskirchen, Germany), and immediately placed in a solution of 20 mM [15]NH$_4$Cl (99% NH$_4$Cl, Campro Scientific GmbH, Berlin, Germany) in a reagent vessel with the leaves still attached to the tree. The twig with the feeding leaves in the experimental set-up was placed in a plastic bag to avoid overly evaporation. The surface of the pots was covered with plastic foil to avoid soil contamination. After 72 h the feeding device and soil cover were removed and the [15]N fed leaves were cut off to finish [15]N uptake. The area of the feeding leaves was scanned and corresponded to a mean of 6% of the total tree leaf area. The latter was determined when the trees were harvested. The labeled leaf areas did not vary among different sets of plants ($P = 0.921$).

The pots with the trees were then placed in plastic bags, which were sealed at the stem bottom to separate the above- and below-ground compartments (Supplementary Fig. S1a, b). To enable irrigation, a tube was introduced into the bag and placed on top of the soil. The prepared trees were transferred into a gas-tight chamber (Weiss Umwelttechnik, Reiskirchen, Germany). The end of the irrigation tube was installed outside of the chamber and used to supply the trees with water. [12]CO$_2$ and [13]CO$_2$ were supplied by silicon tubing to the air space of the gas-tight chamber and circulated with the help of valves and a pumping system (Supplementary Fig. S1c). In the chambers, the trees were exposed for an adjustment phase of 2 days to [12]CO$_2$, then for 3 days to [13]CO$_2$ and then for a chase period of 2 days to [12]CO$_2$ before they were removed. The duration of the chase period was chosen according to Sommer et al.[62] to achieve maximum [13]C enrichment in roots and the labeling intensity was chosen to reach a detectable signal in small amounts of biomass (such as ectomycorrhizas) without any known disturbance of metabolism[63,64]. To produce [13]CO$_2$, 106.98 g Na$_2$[13]CO$_3$ (99 atom-% [13]C, Sigma-Aldrich, Taufkirchen, Germany) was dissolved in 2 L of deionized H$_2$O and automatically mixed with 5 M lactate (90% C$_3$H$_6$O$_3$ pur Ph Eur. USP, AppliChem, Darmstadt, Germany) by a pumping system. During the adjustment phase,

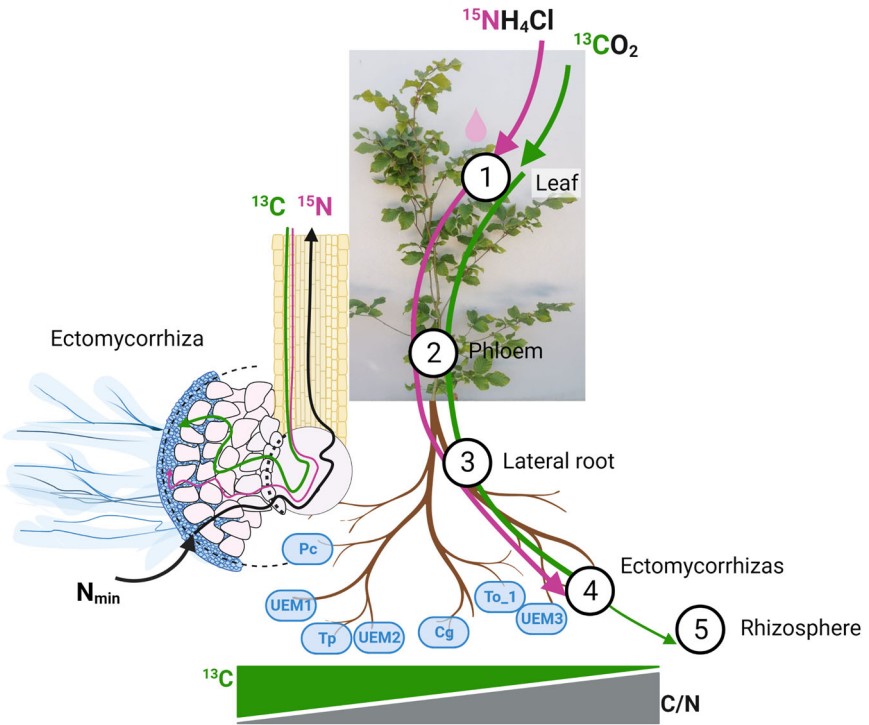

**Fig. 7 The route of leaf-assimilated carbon and nitrogen to ectomycorrhizas formed with different fungal taxa.** The leaf-assimilated carbon (green solid line), nitrogen (pink solid line), ectomycorrhizas formed with different fungal taxa (blue cases). Numbers indicate the compartments where carbon ($^{13}$C) and nitrogen ($^{15}$N) isotope labels applied on the leaf were transported 20 days after labeling application. The left inset represents a cross-section of ectomycorrhiza, illustrating the presence of $^{13}$C and $^{15}$N in both plant and fungal structures. The enrichments with $^{13}$C in the symbiosis decrease with increasing C/N ratio of ectomycorrhizas. A higher fungal nitrogen immobilization from soil ($N_{min}$) may lower the ectomycorrhizal C/N ratio, triggering a higher plant carbon flux to the symbiont. Created with BioRender.com.

the same device was used but supplied with $CO_2$ produced from $Na_2{}^{12}CO_3$ (105.99 g in 2 L deionized $H_2O$, KMF Laborchemie Handes GmbH, Lohmar, Germany) instead. When the $^{12}CO_2$ concentration (senor: Carbocao, GMM220, Driesen+Kern GmbH, Bad Bramstedt, Germany) in the air space dropped below 180 ppm, the pumps were activated and adjusted $^{12}CO_2$ to 400 ppm. The $^{13}CO_2$ concentrations were measured by gas chromatography and reached a maximum of 1800 ppm. $^{13}CO_2$ was applied only during the light phase. During exposure in the chambers, the trees were kept for 16 h under light (420 μmol photons m$^{-2}$ s$^{-1}$ of photosynthetically active radiation) at 70% relative air humidity and 20 °C air temperature. After removal from the chambers, the trees were kept indoors, in ambient air with 16 h/8 h light/dark cycles under the same environmental conditions as in the chambers. The first group of trees was harvested immediately after removal (5 days since the start of the $^{13}CO_2$ treatment) and the second group 15 days later (20 days since the start of the $^{13}CO_2$ treatment). The experimental exposures were conducted in two independent runs, each with 10 trees of which half were harvested in the first and half in the second group (5 or 20 days of $^{13}CO_2$ treatment). Ten non-labeled trees maintained indoors under the same environmental conditions as in the labeling chambers and ambient $CO_2$ (397 ± 4 ppm) were used as controls.

**Harvest and root morphotyping**. The leaves were harvested and scanned to determine plant leaf area. The plant stem was cut off at the soil surface. An ~20 mm long bottom part of the stem was cut and debarked. The bark was dried. The roots with soil were removed from the pots and cautiously shaken. Soil adhering to fine roots was defined as rhizosphere soil. An aliquot was collected with a toothpick (ca. 2 g), and dried. Then the roots were

shaken to remove the remaining soil. The total amount of rhizosphere soil (dry) was 54 ± 10 g per root system. The roots were briefly washed and separated into fine (<2 mm diameter) and coarse roots. Subsamples of coarse and fine roots were dried.

Further subsamples of fresh fine roots were viewed under a stereomicroscope (M 205 FA, Leica, Wetzlar, Germany) and about 400 root tips per plant were classified as vital ectomycorrhizal, vital non-mycorrhizal or dry. Vital and dry root tips were distinguished based on their microscopic appearance, categorizing shrunken and dark brown distorted root tips as dried. This technique was previously validated by stable isotope tracing, demonstrating negligible $^{13}$C or $^{15}$N transfer to the dry root tips of plants supplied via the soil with labeled glucose or ammonium[33,65]. The number of root tips in each category (vital non-mycorrhizal, vital mycorrhizal, dry) was counted and used to calculate:

$$\text{Mycorrhizal colonization rate } (\%) = \text{Number of vital mycorrhizal root tips} * 100 \\ /(\text{number of vital mycorrhizal root tips} \\ + \text{number vital non} - \text{mycorrhizal root tips}).$$

$$\text{Root tip vitality } (\%) = (\text{number of vital mycorrhizal root tips} \\ + \text{number vital non} - \text{mycorrhizal root tips}) * 100 \\ /\text{number of all counted root tips}.$$

Vital ectomycorrhizal root tips were further classified into ectomycorrhizal morphotypes according to fungal characteristics using for instance color, branching, or lengths of emanating hyphae[66] and, subsequently counted to determine the relative abundance of the morphotypes in a sample as follows:

Relative abundance of morphotype$_i$ = number of root tips with morphotype$_i$/number of vital mycorrhizal root tips

Aliquots of each morphotype consisting of fungal and plant tissues were collected (Supplementary Fig. S2). Because individual trees did not contain all morphotypes in sufficient quantities, the number of replicates per morphotype varied ($n = 5$ to $7$). One subsample was stored at $-80\,°C$ for molecular identification, another was fixed in a solution of FAE (formaldehyde 37%:ethanol 70%:acetic acid 97% in proportions of 5:90:5) for Secondary Ion Mass Spectrometry, and the rest sample was dried. The lateral root segment connecting the ectomycorrhizal morphotype with the next higher root branch was also collected[27] (Supplementary Fig. S2a) and dried.

**Identification of morphotypes by internal transcribed spacer (ITS) sequencing.** The DNA of individual morphotypes was extracted in 100 µl lysis buffer of the innuPrep Plant DNA kit (Analytik Jena, Jena, Germany) with a pellet mixer (VWR Pellet Mixer, VWR International, Darmstadt, Germany). Further steps were conducted as described by Lang et al.[3] using the internal transcribed spacer (ITS) region of the fungal rRNA (primers ITS1f (5′CTTGGTCATTTAGAGGAAGTAA-3′) and ITS4 (5′ TCCTCCGCTTATTGATATGC-3′)) and the Fermentas protocol for polymerase EP 0402 (Fermentas, Waltham, Massachusetts, USA). Sequencing was conducted by the company Seqlab (Sequence Laboratories, Göttingen, Germany). The sequences were analyzed with Staden package (http://staden.sourceforge.net) and blasted in NCBI GenBank (www.ncbi.nlm.nih.gov) and UNITE (http://unite.ut.ee) databases. Fungal sequences have been deposited in NCBI GenBank under the accession numbers MK430999 to MK431014.

**Image analyses and secondary ion mass spectrometry (SIMS).** The morphotype samples were dehydrated by incubation in ethanol solutions of increasing concentrations, 80%, 96%, and 100% for 90 min each. Subsequently, the morphotypes were embedded in Technovit 7100 resin (Heraeus Kulzer GmbH & Co. KG, Hanau, Germany) according to the manufacturer´s instructions. Technovit is a well-established acryl plastic embedding system commonly used in NanoSIMS analyses[67,68] because it has negligible N contents. Cross-sections with a thickness of 1.0 µm were cut with an autocut microtome (Ultracut E, Reichert-Jung, Vienna, Austria), stained with toluidine blue[36], and were viewed and photographed under a microscope (Axioplan Observer Z1, Carl Zeiss GmbH, Göttingen, Germany). Digital images of the cross sections were used to measure surface areas of root tissue and fungal tissue with ImageJ (https://imagej.nih.gov). Based on these measurements, we calculated the percentage occupied by fungal structures within ectomycorrhizas as Fungal tissue (%) = area of fungal tissue × 100/total cross-sectional area. Details are shown in Supplementary Table S1. For NanoSIMS imaging, 200 nm thick cross sections were cut using an EM UC6 ultramicrotome (Leica Microsystems, Wetzlar, Germany). The sections were analyzed with a NanoSIMS 50 L instrument (Cameca, Gennevilliers, France) equipped with an 8 kV Cs+ primary ion source[69]. To reach a steady state of ionization, samples were first implanted applying a primary current of ~100 pA for 1.5 minutes. During the analysis, a current of ~1 pA was employed. The detectors were set to collect $^{12}C^{14}N^-$ and $^{12}C^{15}N^-$ secondary ions. The mass resolving power was adjusted to differentiate $^{12}C^{15}N^-$ from potential interferences such as $^{13}C^{14}N^-$. Ion images of $512 \times 512$ pixels were acquired setting a raster size of $30 \times 30$ µm or $80 \times 80$ µm, and a dwell time of 5.07 ms/pixel from three consecutive layers. NanoSIMS data processing, including image exportation, drift correction, image stacking, and ratio measurements was performed using the OpenMIMS plugin of Fiji[70].

**N and C analyses.** For analyses of $^{14}N$, $^{15}N$, $^{12}C$, and $^{13}C$ all dried samples (except morphotypes and lateral roots) were milled and weighed (0.5 to 1.5 mg on a supermicro balance, type S4, Sartorius, Göttingen, Germany) into tin cartouches ($5 \times 9$ mm, HEKAtech, Wegberg, Germany). Each collected morphotype (Supplementary Fig. S2) and lateral root sample was weighed and used as whole sample in the analysis. Since the whole sample was used, no milling was necessary to obtain representative subsamples. The samples of the labeled and the non-labeled controls were analyzed in an isotope ratio mass spectrometer (IRMS; Delta C, Finnigan MAT, Bremen, Germany) in the Center for Stable Isotopes (KOSI, University of Göttingen, Göttingen, Germany). An acetanilide standard (C = 71.09%, N = 10.36%, $\delta^{13}C = -29.6‰$, $\delta^{15}N = -1.6‰$) was measured after each 10th sample and used for point calibration. A second in-house standard (wild boar liver $\delta^{13}C = -17.3‰$, $\delta^{15}N = 7.3$) was used periodically to check for scaling. The standard materials were calibrated against international standards (International Atomic Energy Agency: IAEA-N1 and IAEA-N2 for $^{15}N$ and NBS18 and IAEA 600 for $^{13}C$, https://nucleus.iaea.org/sites/ReferenceMaterials/Pages/Stable-Isotopes.aspx). Atom-percent excess (APE) and the $^{13}C$ enrichment (which is the concentration of newly acquired $^{13}C$) were determined as follows:

$$\text{APE}^{13}C(\%) = \left( \left( \frac{^{13}C}{^{13}C + {}^{12}C} \right)_{labelled} - \left( \frac{^{13}C}{^{13}C + {}^{12}C} \right)_{non-labelled} \right) * 100$$

$$^{13}C \, \text{enrichment} \left( \text{mg g}^{-1} \text{dry mass} \right) = \frac{\text{APE}^{13}C * 1000}{100} * \frac{(^{12}C + {}^{13}C)(g)}{\text{Dry mass}(g)}$$

The amount of newly acquired $^{13}C$ in ectomycorrhizas (per plant) was calculated as

$$\text{Ectomycorrhizal } {}^{13}C \, \text{amount (mg)}$$
$$= \sum_{i=1}^{n} \left( {}^{13}C_{\text{EM species } i} \times \text{root tip biomass}_{\text{EM species } i} \times \text{number of root tips}_{\text{EM species } i} \right)$$

The number of root tips$_{(\text{EM species } i)}$ was calculated as relative abundance$_{(\text{EM species } i)} \times$ total number of root tips per plant. The mean total number of EM tips per plant was $29215 \pm 4361$ ($n = 25$ plants). The root tip biomass$_{(\text{EM species } i)}$ for different EM species is reported in Supplementary Table S1.

The amount of newly acquired $^{13}C$ in roots (per plant) was calculated as

$$\text{Amount of } {}^{13}C \, \text{in roots (mg)} = {}^{13}C_{\text{fineroot}} \times \text{biomass}_{\text{fine root}}$$
$$+ {}^{13}C_{\text{coarse root}} \times \text{biomass}_{\text{coarse root}}$$

APE $^{15}N$, the concentrations of newly acquired $^{15}N$ and the $^{15}N$ amounts were calculated accordingly.

**Statistics and reproducibility.** All source data used for statistical analyses and to produce the graphs and tables have been deposited in Dryad[71]. Statistical analysis was performed using Statgraphics Centurion XVI (StatPoint, Inc., St Louis, MO, USA). To test whether the data were normal distributed, the plots of the residuals were visually inspected. If the data deviated from normal distribution, they were log-transformed and rechecked. The variance was plotted and checked visually for homogeneity prior to one-way or multivariate analysis of variance. Main factors in the ANOVA were time and tissue. When $P \leq 0.05$, a post hoc test was conducted (HSD) to compare means. When time (day 5 and day 20) showed no significant differences at $P \leq 0.05$, the sets of both time points were analyzed together applying general linear models and time as random factor. Data are shown as means ($\pm$SE, $n = 5$ or $10$) when not indicated otherwise. Linear regression curves between $^{13}C$ enrichment in lateral roots and associated EM fungal species were compared using the function "comparison of regression lines". Comparison of the

ectomycorrhizal community composition present at different sampling dates was conducted using the Bray Curtis Similarity Index by ANOSIM in PAST software[72].

**Reporting summary.** Further information on research design is available in the Nature Portfolio Reporting Summary linked to this article.

## Data availability

Fungal sequences have been deposited in NCBI Genbank under the accession numbers MK430999 to MK431014. Source data underlying all graphs and plots in the manuscript are available at Dryad: https://doi.org/10.5061/dryad.k98sf7m6x.

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

## Acknowledgements
This project was funded by the Ministry of Science and Culture of Lower Saxony and the 'Niedersächsisches Vorab' as part of the Cluster of Excellence 'Functional Biodiversity Research' and by the German Research Foundation (DFG) (PE 2256/3-1). We are grateful to the administration of the Göttingen Forest for permission to excavate tree saplings in the Billingshäuser Schlucht. We appreciate the help of T. Klein, B. Kopka, and G. Lehmann (Laboratory for Radio-Isotopes, University of Göttingen) with the labeling procedures in the gas-tight chambers and thank M. Fastenrath for support with anatomical analyses. We thank J. Dyckmans from the Kompetenzzentrum für Stabile Isotope (KOSI, University of Goettingen) for measuring the stable isotopes.

## Author contributions
R.P.: conceptualization, methodology, investigation, formal analysis; S.L.B.: investigation; S.A.: investigation, formal analysis; P.A.G.: formal analyses, investigation; S.O.R.: resources, methodology; S.S.: conceptualization, supervision; A.P.: conceptualization, supervision, formal analysis, funding acquisition; R.P. and A.P.: developed the first draft, all authors: contributed to review and editing the final draft.

## Funding

## Competing interests
The authors declare no competing interests.
