## [Peer Review File · Communications Biology]

Reviewers' comments:

Reviewer #1 (Remarks to the Author):

It is a through conducted study on a very interesting topic. The manuscript is clearly written. There are interesting conclusions expanding the knowledge on the mechanisms relating plant and fungi interactions, and the role C and N of the fungal species play into it. I recommend it for publication. My only concern is the approach used in interpreting the differences between 13C and 15N standardized labels (ll. 311-313) - I do not agree that it is appropriate to ignore the difference in time and interpret the differences in 13C and 15N in such a manner. There were good 5 days gam between the start of 15N and 13C labeling. Of course, a lot could have happened with 15N during that time.

Reviewer #2 (Remarks to the Author):

The manuscript by Pena et al. (Mycorrhizal C/N ratio determines plant-derived carbon and nitrogen allocation to symbiosis) provides a very nice description of potential controls on C and N exchange between beech trees and a variety of fungal taxa in a series of incubation experiments on extracted plants. The manuscript is very well written and includes both a description of the results and nicely leverages these results to make inferences about both the controls and mechanisms of the observed nutrient exchanges (C and N). This is a timely topic and the data presented appear novel and are likely to have a strong impact on the field. I enjoyed reading the manuscript and believe it will appeal to a wide readership. I list below some relatively minor comments related to the text:

- The authors very nicely describe potential mechanisms explaining what they observed as well as alluding to ecosystem-level impacts of the controls on C and N exchange. It may be nice to capture this information in a summary schematic / figure to be included in the discussion section of the manuscript.
- The authors use a nice approach to link different taxonomic groups to the isotopic data and patterns in the C and N exchange with figures 4 and 6 showing some of the taxon-specific results. I recognize that many of the taxonomic groups are uncultured, but can any more be shared on potential different behaviors caused by the different taxa? For instance, the authors indicate in lines, 69-70, 79-82, 366-369, etc. that taxonomic difference in the EC fungi can influence the C and N exchanges. Any more information on this in terms of the data they collected could be informative.
- Figure 5: there is a reference to (line 636) "Stars indicate means for isotopic contents..." but it is unclear what the stars are as I can't seem to locate them in the figure (apologies if I am just missing them).
- Figure 4: The data are nicely labeled with the different taxonomic groups but these labels are obscured at the lower ranges of the plot due to crowding. Perhaps it is possible to more clearly label these points and/or list the data in a supplemental table.
- It was very interesting to track N from tree to EC fungi (as noted in lines 387-388). Do these results shed any light on interactions when the flow of N is reversed?
- Very minor suggested changes:
 - o Lines 382-383: suggest moving "also" to between "roots" and "produce" (to make ...roots also produce and exude other....)
 - o I believe Supplemental Table S2 (line 267) is discussed in the text prior to Supplemental Table S1 (line 288) which suggests they may need to be renamed/ reordered to match the text (unless I missed a previous preference to Sup Table S1)
 - o Line 214: Can the authors confirm they used a "Delta C" IRMS. I am not familiar with this model but know that the "Delta V" is a very common instrument.
 - o Line 177: it may be helpful to list a name/year for publication 3 instead of just inserting a "3" when referring to previous use of the method.

Reviewer #3 (Remarks to the Author):

The authors used stable isotope tracing to investigate C and N flows from beech trees to different components of an ectomycorrhizal system. The experiment is well-designed and the results shed light upon variation in EM acquisition of plant-derived C and N. The results are very interesting and make a valuable contribution to the literature. However, many key details are missing from the manuscript, some of the claims are overstated, and several important sentences are unclear. The manuscript would be much stronger with the addition of more information, more extensive discussion, and revision of several sentences.

58-59: Please provide citations to support this statement.

74-86: These are interesting and reasonable hypotheses. However, the data and interpretation presented in the manuscript are not sufficient to support or refute all of the stated hypotheses. Please narrow down the hypotheses to what is presented in the rest of the manuscript.

74-77: This hypothesis is interesting, but seems too broad. In the experiment described in the manuscript, you focus on C and N allocated from leaves to the EM symbiosis. Please reformulate the hypothesis so that it matches the experiment more specifically.

77-82: What are intrinsic C resources of the EM fungal species? Is this the C content of the existing fungal biomass? Why would EM fungi with high C/N be unlikely to have a high demand for host C? Couldn't the existing fungal biomass C be derived primarily from the host? Is the assumption that different fungal species have different 'optimal' C/N ratios, and that these determine C and N demand from the host? Or that a fungus with low C/N needs more C, and therefore demands more C from the host? Please clarify.

82-83: This sentence is unclear. What is "the demand of the plant fraction in the symbiosis"? Is this total plant N demand? Or N demand of the root tips that the EM colonize? Please clarify and provide citations to support this statement. Or is this a hypothesis that you are testing?

83-85: This sentence is unclear. I think that you intended to communicate that:

- The extent to which different EM species contribute to host nutrition varies substantially across different EM species.
 - Of all possible root activities, nutrient acquisition is the activity that requires the most resources
 - Trees allocate different quantities of N to different ectomycorrhizal species; these differences depend on the extent to which the EM species contributes to host nutrition (and thereby reduces the resources that the plant must invest in root growth and nutrient acquisition)
- However, the way that the sentence is currently written, this is not clear. Additionally, many of these statements should be supported with citations.

95: Please provide more information about the beech trees. For example, report the size, age, and variation between the trees.

127-128: Please explain why the trees were exposed to $^{12}\text{CO}_2$ for 2 days following exposure for ^{13}C and how this might have affected your results.

128: Should this be $\text{Na}^{213}\text{CO}_3$? Also, please specify if the 99% refers to the atom % ^{13}C .

135-136: Are the ppm values correct? Why was $^{12}\text{CO}_2$ adjusted to 400 ppm and $^{13}\text{CO}_2$ adjusted to 1800 ppm? If this is correct, please provide a justification for the different ppm levels and a discussion of how this might have affected your results.

141: Did you measure total plant (leaf, root, stem/branch, etc) or fungal biomass, how much these varied between replicates, and whether there was any correlation between these characteristics and fungal colonization rate, the relative abundance of different fungi, C/N content, etc? These differences could influence your results and should be addressed.

145-146: How were the control/non-labeled trees maintained? Were they also grown in chambers, but only exposed to $^{12}\text{CO}_2$? Or were they grown under different conditions? Please specify. If the control/non-labeled trees were grown under different conditions, please discuss how the difference in conditions did or did not affect the results.

156: Please describe which parameters were used to determine whether root tips were vital.

168: Please describe how morphotypes were collected. Eg, was the whole colonized root tip collected or did you separate fungal tissue from root tissue?

188: Please describe how the morphotypes were harvested/preserved/stored prior to embedding, and how this as well as sample preparation for NanoSIMS might affect C/N content.

200-201, Fig S3: Which masses did you collect data from? Did you calculate ^{15}N and ^{14}N based on $^{12}\text{C}^{15}\text{N}$ - and $^{13}\text{C}^{14}\text{N}$ - alone, or did you collect the $^{12}\text{C}^{14}\text{N}$ - image, too? Did you collect data from any other masses? How many regions did you image? Please provide more information.

211: Please specify whether the morphotypes included root and fungal tissue, or only fungal tissue.

244: Please remove the Δ sign. I think that it could be misleading, since you report the isotope values as APE rather than in delta notation.

245: Please replace "signature" with "enrichment"

244-247: Did you quantify total fungal, plant, and rhizosphere soil mass? Can you convert the ^{13}C , ^{15}N , total C, and total N concentrations to a total mass value? This seems necessary in order to discuss C and N allocation to different components.

260-261: Please describe how you calculated the proportion of fungal tissue in EM root tips. Did you separate fungal tissue from root tissue, measure the dry mass separately, and calculate the proportion of fungal:total mass? Or did you estimate the proportion of fungal tissue through image analysis? Or does the percentage indicate the percent colonization of vital root tips (as described in lines 160-161)?

272: The sink strength/size depends both on ^{13}C APE and on total fungal mass, which may not be positively correlated in every case. Can you calculate and report total sink size? I think that would be a very interesting addition to the manuscript. If not, this statement needs to be qualified. Also, please discuss how ^{13}C allocation and enrichment might vary depending on tree size, age, and seasonal conditions.

275, 280, etc: I think that it is misleading to discuss the ^{13}C and ^{15}N trends in terms of "accumulation," since you do not report total mass allocated to different components. Please revise.

280-283: Is this the only observation worth noting from the NanoSIMS imaging? Did you collect images from any regions other than the ones presented in the SI? It is not clear why the NanoSIMS data were collected and what they help to explain about EM C/N dynamics.

287-288: Does the "proportion of fungal tissue" refer to the mass, % colonization, or something else? Please elaborate.

295-298: This is an interesting idea. As indicated in my comments above, I'm not certain whether your ectomycorrhizal data are based on fungal tissue alone or combined fungal and root tip tissue. If it is the latter, it is difficult to use bulk IRMS data to determine whether elemental/isotopic differences in symbiotic tissues are due to differences in fungal content, root tissue content, or both. Can you use your NanoSIMS data to determine the elemental/isotopic composition of fungal and root tissues within symbiotic root tips, and further support your hypothesis?

307-308: Please provide a justification for using the bark rather than the stem or whole aboveground plant tissue as the material for normalization.

320-321: This sentence is unclear. Please revise.

321-322: This could be due to the larger starting C pool in high C/N tissues. If the same quantity of new ¹³C is allocated to a high C tissue and a low C tissue, it will be more diluted in the high C tissue. Please address this in your interpretation.

335-336: I do not think that there is sufficient evidence for this statement. Please elaborate or revise.

337: Without total mass values, I do not think that it is possible to know which fungi received more C. Please revise.

334-354: The ideas in these paragraphs are very interesting, but not sufficiently developed/supported. Please expand.

362-363: Please provide a more detailed explanation of what you mean by "limits" and how these limits are determined by fungal C/N ratio. It is not clear how you can test this hypothesis with your data, since you did not impose different nutrient conditions.

364-366: This text is unclear. What does it mean that "C allocation to EM root tips is not compensated by corresponding N immobilization"? Are you referring to fungal immobilization of plant-derived N? Who is compensating whom, and for what? Why would C allocation to root tips be compensated by fungal N immobilization?

364-369: In order to discuss mycorrhizal N dynamics, it seems necessary to consider N that the fungus derives directly from the soil, and to consider longer temporal dynamics of resource exchange between the fungus and plant. Is it possible to expand your data analysis to include this?

373-375: The logic in this sentence is not clear. Is it implied that allocation of host N to root tips is indicative of internal N availability, and additional N uptake is low when internal N availability is high, and therefore if N allocation to root tips is high, additional N uptake will be low? Please clarify.

378-380: This sentence is unclear. Are you trying to say that 10% of the NH₄Cl that was applied to the leaves was allocated to the mycorrhizal root tips? Please revise.

380-385: The main point of these sentences is unclear. The mechanism of N transfer between plants and fungi is very interesting, but your experimental design and results are not suited to investigate this. I do not think that it is necessary to speculate about the mechanism of transfer, unless you expand other data interpretation related to this. Alternatively, these sentences could be combined with the paragraph below. Either way—it is not clear whether the speculation about mechanisms improves the manuscript.

388-389: N allocation and colonization likely occurred on very different timelines in this study. Therefore, I am not convinced that it is possible to use the results presented here to assess whether N allocation to EM played a role in colonization intensity. Please revise.

391-396: These sentences are unclear. Please revise.

397-405: Some of the conclusions are overstated. For example, although EM C/N ratios were correlated with C and N allocation patterns, I do not think that this is sufficient to state that EM C/N ratios drive C and N allocation. Please revise.

Fig 4 and Fig 6: Would it be possible to indicate the fungal species in different colors or shapes instead of (or in addition to) numbers?

Fig 5: I do not see the star symbols indicated in line 636

Table S1: How was the fraction of fungal tissue estimated based on cross sections? Please elaborate in the methods.

Fig S3: It is difficult to tell which regions represent fungal vs plant tissue. Please draw a border to distinguish between the two, or provide another way to distinguish between them.

We thank the three reviewers for their time and insightful comments, which helped us to improve our paper. The lines numbers refer to the approximate position in the “clean” version. We also provide a track-changes-version, where you can view the changes.

Referee #1

Comments

It is a through conducted study on a very interesting topic. The manuscript is clearly written. There are interesting conclusions expanding the knowledge on the mechanisms relating plant and fungi interactions, and the role C and N of the fungal species play into it. I recommend it for publication.

Thank you. We are glad to read that this reviewer found our study interesting and clearly written and recommended it for publication.

My only concern is the approach used in interpreting the differences between ^{13}C and ^{15}N standardized labels (ll. 311-313) - I do not agree that it is appropriate to ignore the difference in time and interpret the differences in ^{13}C and ^{15}N in such a manner. There were good 5 days gam between the start of ^{15}N and ^{13}C labeling. Of course, a lot could have happened with ^{15}N during that time.

This is a very good point. We understand the concerns of this reviewer because we did not apply the ^{15}N and ^{13}C labeling simultaneously, which may undermine a direct calculation of a ratio of C and N allocation. We have now removed this analysis. Since the question "how much N is allocated to roots compared C" is obvious, we include in the Discussion (lines 444-449): "An obvious question is how much N is allocated to ectomycorrhizas compared with C. Based on our data, we cannot provide a conclusive answer because of the time lapse between the ^{15}N and the ^{13}C labelling and the differences in label intensities. However, comparing the $^{13}\text{C}/^{15}\text{N}$ ratios, which increased from 259 in roots to 3043 in ectomycorrhizas, in our experiment the plant provided 11 units of ^{13}C for each unit of ^{15}N to the EM root tips. Further studies are required to refine this analysis."

Referee #2

Comments

The manuscript by Pena et al. (Mycorrhizal C/N ratio determines plant-derived carbon and nitrogen allocation to symbiosis) provides a very nice description of potential controls on C and N exchange between beech trees and a variety of fungal taxa in a series of incubation experiments on extracted plants. The manuscript is very well written and includes both a description of the results and nicely leverages these results to make inferences about both the controls and mechanisms of the observed nutrient exchanges (C and N). This is a timely topic and the data presented appear novel and are likely to have a strong impact on the field. I enjoyed reading the manuscript and believe it will

appeal to a wide readership. I list below some relatively minor comments related to the text.

Thanks a lot for your positive and constructive comments. We are glad to read that this reviewer found our paper timely, well-written, and our methods to be detailed.

The authors very nicely describe potential mechanisms explaining what they observed as well as alluding to ecosystem-level impacts of the controls on C and N exchange. It may be nice to capture this information in a summary schematic / figure to be included in the discussion section of the manuscript.

This is an excellent suggestion which strengthens our findings. We have now included a summary picture Fig. 7.

The authors use a nice approach to link different taxonomic groups to the isotopic data and patterns in the C and N exchange with figures 4 and 6 showing some of the taxon-specific results. I recognize that many of the taxonomic groups are uncultured, but can any more be shared on potential different behaviors caused by the different taxa? For instance, the authors indicate in lines, 69-70, 79-82, 366-369, etc. that taxonomic difference in the EC fungi can influence the C and N exchanges. Any more information on this in terms of the data they collected could be informative.

This is an important comment. We have now cited a couple of references to highlight a comparison among taxa of our study (lines 420-423): "In line with this idea, ectomycorrhizas formed with *C. geophilum*, which exhibit lower N acquisition abilities than those with *P. conglomerata*, some Helotiales, or *T. punicea*^{31,32,33}, showed a low ¹³C enrichment and a high C/N ratio in the present study."

Figure 5: there is a reference to (line 636) "Stars indicate means for isotopic contents..." but it is unclear what the stars are as I can't seem to locate them in the figure (apologies if I am just missing them).

Thank you for highlighting this. We have now modified the caption of Figure 5.

Figure 4: The data are nicely labeled with the different taxonomic groups but these labels are obscured at the lower ranges of the plot due to crowding. Perhaps it is possible to more clearly label these points and/or list the data in a supplemental table.

Thank you. We enhanced the clarity of Figure 4 by incorporating coloured symbols.

It was very interesting to track N from tree to EC fungi (as noted in lines 387-388). Do these results shed any light on interactions when the flow of N is reversed?

We are not sure that we understood this question correctly. In plants, N is normally transported bidirectionally (see line 430) from the soil via the xylem to the leaves and from the leaves via the phloem (localized in the bark) to other sink tissues. In this sense, there will be no flux reversal but probably a mixing of N sources in ectomycorrhizas from fungal uptake and plant N circulation. With the isotope technologies available to us, these sources cannot be directly discerned because we can either label the soil N or the plant N but not both simultaneously.

Very minor suggested changes:

Lines 382-383: suggest moving "also" to between "roots" and "produce" (to make ...roots also produce and exude other....)

Done.

I believe Supplemental Table S2 (line 267) is discussed in the text prior to Supplemental Table S1 (line 288) which suggests they may need to be renamed/ reordered to match the text (unless I missed a previous preference to Sup Table S1)

Thank you. Table S1 has been previously mentioned in the text (line 256).

Line 214: Can the authors confirm they used a "Delta C" IRMS. I am not familiar with this model but know that the "Delta V" is a very common instrument.

We have confirmed with our colleagues from the Center for Stable Isotopes that they used a Delta C spectrometer, as noted under Methods (line 242). Delta C is preferred when strongly labelled samples are used.

Line 177: it may be helpful to list a name/year for publication 3 instead of just inserting a "3" when referring to previous use of the method.

Done. Thank you.

Referee #3

Comments

The authors used stable isotope tracing to investigate C and N flows from beech trees to different components of an ectomycorrhizal system. The experiment is well-designed and the results shed light upon variation in EM acquisition of plant-derived C and N. The results are very interesting and make a valuable contribution to the literature. However, many key details are missing from the manuscript, some of the claims are overstated, and several important sentences are unclear. The manuscript would be much stronger with the addition of more information, more extensive discussion, and revision of several sentences.

Many thanks for your positive and thoughtful comments on our manuscript. We feel that these comments have significantly improved the quality of our manuscript. We address all of them below.

58-59: Please provide citations to support this statement.

Done (line 60): reference ²⁸, Taylor and Alexander 2005. The ectomycorrhizal symbiosis: life in the real world. *Mycologist* 19 (3) 102-112.

74-86: These are interesting and reasonable hypotheses. However, the data and interpretation presented in the manuscript are not sufficient to support or refute all of the stated hypotheses. Please narrow down the hypotheses to what is presented in the rest of the manuscript.

According to your suggestion, we clarified our reasoning and specified our hypotheses (lines 71-86): "The main objective of this study was to examine nutrient concentrations and to determine the sink activity for host resources of ectomycorrhizas. We investigated the incorporation of C and N originating from aboveground leaf assimilation into ectomycorrhizas formed with different fungal taxa in naturally composed assemblages. Since N and C metabolism are linked, and N accumulation varies profoundly among different EM fungal species ^{27,32-34}, we reasoned that plant C transfer to an EM fungal species is determined not only by fungal biomass ²⁶ but also by fungal taxon-specific N utilization. Hence, C provision to an ectomycorrhiza would be related to its taxon-specific C/N stoichiometry. We hypothesized that an EM fungal species with a high C and low N content would be unlikely to have a high demand for host C in contrast to an EM fungal species with low C and high N content. In contrast to C, we expected that host N allocation to ectomycorrhizas reflects the demand of the root cells in the symbiosis and, therefore, would not vary in ectomycorrhizas formed with different fungal species. However, given the large variation among EM species in N acquisition ³⁴ and the unknown impact on the N content in the root cells in the symbiosis, we also considered the alternative hypothesis that tree-derived N transfer to ectomycorrhizas is affected by EM fungal species."

74-77: This hypothesis is interesting, but seems too broad. In the experiment described in the manuscript, you focus on C and N allocated from leaves to the EM symbiosis. Please reformulate the hypothesis so that it matches the experiment more specifically.

Please see the previous answer.

77-82: What are intrinsic C resources of the EM fungal species? Is this the C content of the existing fungal biomass? Why would EM fungi with high C/N be unlikely to have a high demand for host C? Couldn't the existing fungal biomass C be derived primarily from the host? Is the assumption that different fungal species have different 'optimal' C/N ratios, and that these determine C and N demand from the host? Or that a fungus with low C/N needs more C, and therefore demands more C from the host? Please clarify.

We agree that our reasoning was weak here. We have now modified this text and avoided unclear terms as outlined in the previous answer.

82-83: This sentence is unclear. What is "the demand of the plant fraction in the symbiosis"? Is this total plant N demand? Or N demand of the root tips that the EM colonize? Please clarify and provide citations to support this statement. Or is this a hypothesis that you are testing?

We agree and have rephrased the sentence for clarity. Please see above.

95: Please provide more information about the beech trees. For example, report the size, age, and variation between the trees.

We now include more detailed information in lines 105-107. " Twenty trees of similar size (\pm SE) were selected for labelling: 133 ± 4 cm height, 12.05 ± 0.29 mm width at the stem base, 0.23 ± 0.02 m² whole-plant leaf surface area, 7.3 ± 0.6 g leaf biomass, 12.5 ± 1.0 g coarse root and 3.9 ± 1.1 g fine root biomass"

127-128: Please explain why the trees were exposed to 12CO2 for 2 days following exposure for 13C and how this might have affected your results.

We have now inserted (lines 129-131): "...then for a chase period of two days to ¹²CO₂ before they were removed. The duration of the chase period was chosen according to ³⁷ to achieve maximum ¹³C enrichment in roots...."

128: Should this be Na²¹³CO₃? Also, please specify if the 99% refers to the atom % ¹³C.

Thank you. Corrected.

135-136: Are the ppm values correct? Why was 12CO2 adjusted to 400 ppm and 13CO2 adjusted to 1800 ppm? If this is correct, please provide a justification for the different ppm levels and a discussion of how this might have affected your results.

We understand the concern of this reviewer and agree that long-term exposure of trees to CO₂ concentrations well above ambient levels may alter tree metabolism. However, here we used short-term pulse labeling under conditions which have no known impact compared with long-term labeling using ambient CO₂ concentrations (Epron *et al.*, 2012). 400 ppm is the ambient CO₂ concentration. The elevated CO₂ concentration during ¹³C pulse labeling results in a significant ¹³C label, which can be traced in different plant compartments and in the soil. It is important to note that mycorrhizas have only minute mass and, therefore, the label has to be sufficiently high to be traceable in a sample of about 500 μ g. To clarify, we have inserted (lines 130-133): "The duration of the chase period was chosen according to ³⁷ to achieve maximum ¹³C enrichment in

roots and the labelling intensity was chosen to reach a detectable signal in small amounts of biomass (such as ectomycorrhizas) without any known disturbance of metabolism^{38,39}."

141: Did you measure total plant (leaf, root, stem/branch, etc) or fungal biomass, how much these varied between replicates, and whether there was any correlation between these characteristics and fungal colonization rate, the relative abundance of different fungi, C/N content, etc? These differences could influence your results and should be addressed.

This is a very good point. However, we selected plants of similar size (as noted above) and thus, the range of the data was not sufficiently large for such an analysis.

145-146: How were the control/non-labeled trees maintained? Were they also grown in chambers, but only exposed to 12CO₂? Or were they grown under different conditions? Please specify. If the control/non-labeled trees were grown under different conditions, please discuss how the difference in conditions did or did not affect the results.

We have clarified this point in lines 151-153 "Ten non-labeled trees maintained indoors under the same light and temperature conditions as in the labeling chambers and at ambient CO₂ (397 ± 4 ppm) were used as controls."

We understand the concern of this reviewer about the difference in C availability between labeled trees exposed to enhanced C concentration during the pulse labeling and control trees. The pulse labeling involves a short exposure to elevated CO₂ that is followed by exposure to ambient CO₂ concentrations. No differences in C allocation or partitioning between plant and soil compartments were apparent between trees ¹³C-pulsed labeled grown under elevated CO₂ during the labeling and ¹³C-continuous labeled grown under ambient CO₂ (Studer *et al.*, 2014).

156: Please describe which parameters were used to determine whether root tips were vital.

We have now included more detail on the separation of the two root tip categories (lines 166-170): "Vital and dry root tips were distinguished based on their microscopic appearance, categorizing shrunken and dark brown distorted root tips as dried. This technique was previously validated by stable isotope tracing, demonstrating negligible ¹³C or ¹⁵N transfer to the dry root tips of plants supplied via the soil with labelled glucose or ammonium^{33,40}."

168: Please describe how morphotypes were collected. Eg, was the whole colonized root tip collected or did you separate fungal tissue from root tissue?

This information is presented in Supplement Fig. S2 (Typical collection scheme for ectomycorrhizas and the attached lateral root segment) and highlighted in the text (lines 185-186): "Aliquots of each morphotype consisting of fungal and plant tissues were collected (Supplement Fig. S2)."

188: Please describe how the morphotypes were harvested/preserved/stored prior to embedding, and how this as well as sample preparation for NanoSIMS might affect C/N content.

We understand the concern of this reviewer that sample preparation (i.e., embedding) may contribute significant amounts of ^{14}N that dilute the isotopic signal. We would like to emphasize that Technovit is a well-established acryl plastic embedding system commonly used in NanoSIMS analyses (McGlynn *et al.*, 2015; Metcalfe *et al.*, 2021) because it has negligible N contents. Calibration could be required to correct for isotopic C and H (Kopf *et al.*, 2016). In our study, we did not employ NanoSIMS as a quantitative method. Instead, we compared the ^{15}N -labeled vs non-labeled samples to visualize the transfer of labeled N to mycorrhizal fungal structures.

We have now added the details on sample handling before embedding: "One subsample was stored at $-80\text{ }^{\circ}\text{C}$ for molecular identification, another was fixed in a solution of FAE (formaldehyde 37%: ethanol 70%: acetic acid 97% in proportions of 5: 90: 5) for Secondary Ion Mass Spectrometry, and the rest sample was dried" (lines 208-216). "The morphotype samples were dehydrated by incubation in ethanol solutions of increasing concentrations, 80%, 96% and 100% for 90 min each. Subsequently, the morphotypes were embedded in Technovit 7100 resin (Heraeus Kulzer GmbH & Co. KG, Hanau, Germany) according to the manufacturer's instructions. Technovit is a well-established acryl plastic embedding system commonly used in NanoSIMS analyses ^{42,43} because it has negligible N contents. Cross-sections with a thickness of $1.0\text{ }\mu\text{m}$ were cut with an autocut microtome (Ultracut E, Reichert-Jung, Vienna, Austria), stained with toluidine blue ⁴⁴, and were viewed, and photographed under a microscope (Axioplan Observer Z1, Carl Zeiss GmbH, Göttingen, Germany).

200-201, Fig S3: Which masses did you collect data from? Did you calculate ^{15}N and ^{14}N based on $^{12}\text{C}^{15}\text{N}$ - and $^{13}\text{C}^{14}\text{N}$ - alone, or did you collect the $^{12}\text{C}^{14}\text{N}$ - image, too? Did you collect data from any other masses? How many regions did you image? Please provide more information.

We have now included (lines 227-228): "The detectors were set to collect $^{12}\text{C}^{14}\text{N}^-$ and $^{12}\text{C}^{15}\text{N}^-$ secondary ions."

NanoSimS studies of ^{13}C in EM tissues have already been published by others (Mayrhofer *et al.* 2021), and therefore, we focused on questions referring to N transfer. We used $^{12}\text{C}^{14}\text{N}$, $^{12}\text{C}^{15}\text{N}$, and the ratio image obtained by dividing $^{12}\text{C}^{15}\text{N}/^{12}\text{C}^{14}\text{N}$. Other masses (^{16}O , ^{31}P , and ^{32}S) were not relevant. We imaged a total of two further morphotypes, both yielding similar results, and present here results for *P. conglomerata*, which was the most abundant fungal species in our samples.

211: Please specify whether the morphotypes included root and fungal tissue, or only fungal tissue.

We have now presented this information in lines (185-186). For clarity, we also included a reference to Supplement Fig. S2, which depicts a morphotype and how it was collected.

244: Please remove the Δ sign. I think that it could be misleading, since you report the isotope values as APE rather than in delta notation.

Done.

245: Please replace "signature" with "enrichment"

Done

244-247: Did you quantify total fungal, plant, and rhizosphere soil mass? Can you convert the ^{13}C , ^{15}N , total C, and total N concentrations to a total mass value? This seems necessary in order to discuss C and N allocation to different components.

Thank you. This is an important aspect in discussing our results and we have included the relevant information for roots, ectomycorrhizas and rhizosphere soil in the Methods (lines 250-260), additional supplementary information in Table S1 and Figure S3, and results (lines 284-287 and lines 291-292): "The total pools of newly acquired ^{13}C and ^{15}N in roots were $154.1 \pm 18.8 \text{ mg } ^{13}\text{C}$ and $142.2 \pm 29.2 \mu\text{g } ^{15}\text{N}$ (means across both sampling dates of $n = 10$ plants \pm SE). The share of ^{13}C and ^{15}N contributed by fine roots to the total root pools were $27.1 \pm 4.4 \text{ mg } ^{13}\text{C}$ and $23.3 \pm 4.2 \mu\text{g } ^{15}\text{N}$ ""The pool of rhizosphere ^{13}C was $1.9 \pm 0.3 \text{ mg}$ per root system (means across both sampling dates of $n = 10$ plants \pm SE)."

260-261: Please describe how you calculated the proportion of fungal tissue in EM root tips. Did you separate fungal tissue from root tissue, measure the dry mass separately, and calculate the proportion of fungal:total mass? Or did you estimate the proportion of fungal tissue through image analysis? Or does the percentage indicate the percent colonization of vital root tips (as described in lines 160-161)?

This has now been clarified by referring to the Supplement Table S1, where we further explained in the legend that the image analysis was used and under Methods (lines 218-220): "Based on these measurements, we calculated the percentage occupied by fungal structures within ectomycorrhizas as

Fungal tissue (%) = area of fungal tissue x 100 / total ectomycorrhizal cross sectional area."

272: The sink strength/size depends both on ^{13}C APE and on total fungal mass, which may not be positively correlated in every case. Can you calculate and report total sink size? I think that would be a very interesting addition to the manuscript. If not, this statement needs to be qualified. Also, please discuss how ^{13}C allocation and enrichment might vary depending on tree size, age, and seasonal conditions.

This is a good point which can only strengthen our findings. We have now included this information (lines 351-354): "Based on the abundance and biomass of the EM species and their enrichments with ^{13}C and ^{15}N , we estimated the EM pool sizes of the transferred ^{13}C and ^{15}N as $1.59 \pm 0.19 \text{ mg } ^{13}\text{C}$ and $0.52 \pm 0.08 \mu\text{g } ^{15}\text{N}$ (means of $n = 20$ plants). The contribution of the individual EM species to the total EM fungal pool size is shown in Supplement Fig. S3A."

We extended the discussion (lines 390-400): "We found that approximately 6% of the new ^{13}C present in fine roots was located in the EM fraction. Assuming that all ^{13}C found in the rhizosphere also passed through the ectomycorrhizas, approximately 13% of the ^{13}C in fine roots was allocated to ectomycorrhizas and their associated mycorrhizosphere. These fractions are expected to vary with the exposure time and are also likely affected by tree phenology because the direction of C flux depends on sink activities such as leaf and wood production in spring and summer, and nutrient resorption and allocation to storage pools in fall^{53,54}. In our study, which was conducted after the main phase of leaf and wood production, coarse roots were a great carbon sink, containing approximately 5 times more ^{13}C than fine roots. Carbon stored in coarse roots can be used to supply ectomycorrhizas when recent photoassimilates are unavailable¹⁸."

275, 280, etc: I think that it is misleading to discuss the ^{13}C and ^{15}N trends in terms of "accumulation," since you do not report total mass allocated to different components. Please revise.

We agree that "accumulation" is inadequate as we have not referred to biomass content. We have now checked throughout the manuscript, replacing "accumulation" with "transfer" or "enrichment".

280-283: Is this the only observation worth noting from the NanoSIMS imaging? Did you collect images from any regions other than the ones presented in the SI? It is not clear why the NanoSIMS data were collected and what they help to explain about EM C/N dynamics

We employed NanoSIMS which is a very costly technique with the specific aim of visualizing the presence of ^{15}N into the ectomycorrhizal fungal structures. We believe this is an essential addition to our study as it demonstrates for the first time to our best knowledge that plant-derived N reaches the fungal structures in the symbiosis. The results also show that the N flux is not unidirectional from the fungus to the plant but also from the plant to the fungus. NanoSims yielded an important qualitative result.

Quantitative data on the C/N dynamics were obtained by measuring allocation of ^{13}C and ^{15}N to fungi and roots.

287-288: Does the "proportion of fungal tissue" refer to the mass, % colonization, or something else? Please elaborate.

We have now clarified this point (lines 332-334) " to the proportion of fungal biomass in the symbioses (Supplement Table S1)."

295-298: This is an interesting idea. As indicated in my comments above, I'm not certain whether your ectomycorrhizal data are based on fungal tissue alone or combined fungal and root tip tissue. If it is the latter, it is difficult to use bulk IRMS data to determine whether elemental/isotopic differences in symbiotic tissues are due to differences in fungal content, root tissue content, or both. Can you use your NanoSIMS data to determine the elemental/isotopic composition of fungal and root tissues within symbiotic root tips, and further support your hypothesis?

This is a very important point. We agree that we cannot separate the isotope enrichments between fungus and plant tissues in the ectomycorrhiza. Our results showed the relationships between ^{13}C or ^{15}N enrichments in ectomycorrhiza and the attached lateral roots. As the plant component is the same in all ectomycorrhiza taxa, we reckon that the variation is given by the fungal species.

307-308: Please provide a justification for using the bark rather than the stem or whole aboveground plant tissue as the material for normalization.

These lines have been deleted. However, the justification was that the phloem in the bark is translocating C and N to the roots and therefore, variations in ^{13}C and ^{15}N in the bark were assumed to reflect variation in transport to the roots.

320-321: This sentence is unclear. Please revise.

Thank you. We noted the confusion because we used "EM fungal species" while the measurements were carried out on ectomycorrhiza species". Given that at the reviewer's suggestion, we calculated the ^{13}C and ^{15}N contents in fungal biomass and proved that the relations between isotope enrichments and C/N ratio have not changed, we are confident the terminology is corrected. However, we have now modified the sentence to avoid misunderstandings (lines 360-361).

321-322: This could be due to the larger starting C pool in high C/N tissues. If the same quantity of new ^{13}C is allocated to a high C tissue and a low C tissue, it will be more diluted in the high C tissue. Please address this in your interpretation.

This is an important comment. We have now addressed the concerns of this reviewer by calculating the ^{13}C and ^{15}N contents in the fungal biomass and proving that the relationship with the C/N ratio has not changed. We have now included these data in the Supplement Fig. S3B, C.

335-336: I do not think that there is sufficient evidence for this statement. Please elaborate or revise.

We have refined the used term ("ectomycorrhiza" instead of "fungal") to avoid misunderstandings of what was exactly measured. However, as per reviewer's suggestions, the new calculations have strengthened our findings and the statement is now correct.

337: Without total mass values, I do not think that it is possible to know which fungi received more C. Please revise.

Agree. We have included the required information (lines 345-355)" The contribution of the individual EM species to the total EM fungal pool size is shown in Supplement Fig. S3A." Given the addition, we have not rephrased this part of the Discussion.

334-354: The ideas in these paragraphs are very interesting, but not sufficiently developed/supported. Please expand.

We have now provided a better explanation (lines 391-394 "We found that approximately 6% of the new ^{13}C present in fine roots was located in the EM fraction. Assuming that all ^{13}C found in the rhizosphere also passed through the ectomycorrhizas, approximately 13% of the ^{13}C in fine roots was allocated to ectomycorrhizas and their associated mycorrhizosphere."

362-363: Please provide a more detailed explanation of what you mean by "limits" and how these limits are determined by fungal C/N ratio. It is not clear how you can test this hypothesis with your data, since you did not impose different nutrient conditions.

We have clarified (lines 417-421): "... because under our experimental conditions, the ^{13}C allocation to ectomycorrhizas was almost zero when their C/N ratio was greater than 24. Our results imply that C allocation to ectomycorrhizas, which is not accompanied by N uptake from soil and N immobilization in the fungus, would lead to an increase in the EM C/N ratio and consequently may result in reduced C flux to the symbiont."

364-366: This text is unclear. What does it mean that "C allocation to EM root tips is not compensated by corresponding N immobilization"? Are you referring to fungal

immobilization of plant-derived N? Who is compensating whom, and for what? Why would C allocation to root tips be compensated by fungal N immobilization?

We have rephrased the sentence as previously described.

364-369: In order to discuss mycorrhizal N dynamics, it seems necessary to consider N that the fungus derives directly from the soil, and to consider longer temporal dynamics of resource exchange between the fungus and plant. Is it possible to expand your data analysis to include this?

We agree that this would be the next step but beyond scope of the present goals. Here, we clearly used the term "we speculate" and the next phrase was expanded to refer explicitly to this uncertainty (lines 425-428): "Further investigation is required to examine the extent to which these fungal traits may vary during longer temporal trajectories across different environments"

373-375: The logic in this sentence is not clear. Is it implied that allocation of host N to root tips is indicative of internal N availability, and additional N uptake is low when internal N availability is high, and therefore if N allocation to root tips is high, additional N uptake will be low? Please clarify.

Yes, that is the message. We have rephrased (lines 431-435): "Trees circulate N bidirectionally between roots and leaves in xylem and phloem, thereby enabling distribution of N at the whole-tree level between storage pools (mainly stem and coarse roots in deciduous trees) and active growth processes⁶³. Increasing cellular N pools, especially of amino acids, suppress N transport systems and, thus, regulate tree N uptake⁶³."

378-380: This sentence is unclear. Are you trying to say that 10% of the NH₄Cl that was applied to the leaves was allocated to the mycorrhizal root tips? Please revise.

We have removed the sentence and the relationship between the allocated ¹³C and ¹⁵N was phrased more carefully (lines 445-450), following the advice of another reviewer.

380-385: The main point of these sentences is unclear. The mechanism of N transfer between plants and fungi is very interesting, but your experimental design and results are not suited to investigate this. I do not think that it is necessary to speculate about the mechanism of transfer, unless you expand other data interpretation related to this. Alternatively, these sentences could be combined with the paragraph below. Either way—it is not clear whether the speculation about mechanisms improves the manuscript.

Our study revealed that plant-derived N occurred in fungal structures of ectomycorrhizas. The novelty of these findings requires at least a very short discussion

on the possible mechanisms involved in them (here four sentences). We believe this is particularly important for a broader audience that includes plant physiologists.

388-389: N allocation and colonization likely occurred on very different timelines in this study. Therefore, I am not convinced that it is possible to use the results presented here to assess whether N allocation to EM played a role in colonization intensity. Please revise.

We agree that the extrapolation of a short study results to a broader scale requires caution. We have rephrased (lines 453-454): "In contrast to C, we found no relationship of plant N allocation to EM root tips and fungal colonization intensity."

391-396: These sentences are unclear. Please revise.

Thank you. We have rewritten and clarified the sentences (lines 455-458). "EM fungi possess several amino acid transporters ⁶¹, implying that they have the potential to recapture the plant-exuded N. The absence of a significant ¹⁵N enrichment in the rhizosphere may suggest that N was retained within plant-fungal symbiosis."

397-405: Some of the conclusions are overstated. For example, although EM C/N ratios were correlated with C and N allocation patterns, I do not think that this is sufficient to state that EM C/N ratios drive C and N allocation. Please revise.

We agree and revised (lines 462-463): In conclusion, we show that the C/N ratios of EM fungal species were correlated with plant-derived C and N allocation to root tips (Fig. 7).

Fig 4 and Fig 6: Would it be possible to indicate the fungal species in different colors or shapes instead of (or in addition to) numbers?

This is a great suggestion. We have now used coloured symbols.

Fig 5: I do not see the star symbols indicated in line 636

The reference to the star symbol has been deleted from this caption.

Table S1: How was the fraction of fungal tissue estimated based on cross sections? Please elaborate in the methods.

Thank you. We have now added more info in the Methods (lines 217-220).

Fig S3: It is difficult to tell which regions represent fungal vs plant tissue. Please draw a border to distinguish between the two, or provide another way to distinguish between them.

Agree. We have now added a dotted line separating the two tissues.

References

- Epron D, Bahn M, Derrien D, Lattanzi FA, Pumpanen J, Gessler A, Högberg P, Maillard P, Dannoura M, Gérant D, *et al.* 2012. Pulse-labelling trees to study carbon allocation dynamics: a review of methods, current knowledge and future prospects. *Tree Physiology* 32: 776–798.
- Kopf SH, Sessions AL, Cowley ES, Reyes C, Van Sambeek L, Hu Y, Orphan VJ, Kato R, Newman DK. 2016. Trace incorporation of heavy water reveals slow and heterogeneous pathogen growth rates in cystic fibrosis sputum. *Proceedings of the National Academy of Sciences* 113: E110–E116.
- McGlynn SE, Chadwick GL, Kempes CP, Orphan VJ. 2015. Single cell activity reveals direct electron transfer in methanotrophic consortia. *Nature* 526: 531–535.
- Metcalf KS, Murali R, Mullin SW, Connon SA, Orphan VJ. 2021. Experimentally-validated correlation analysis reveals new anaerobic methane oxidation partnerships with consortium-level heterogeneity in diazotrophy. *The ISME Journal* 15: 377–396.
- Sommer J, Dippold M, Zieger SL, Handke A, Scheu S, Kuzyakov Y. 2017. The tree species matters: Belowground carbon input and utilization in the myco-rhizosphere.
- Studer MS, Siegwolf RTW, Abiven S. 2014. Carbon transfer, partitioning and residence time in the plant-soil system: a comparison of two $^{13}\text{CO}_2$ labelling techniques. *Biogeosciences* 11: 1637–1648.

Reviewers' comments:

Reviewer #2 (Remarks to the Author):

Please see attached PDF file.

Reviewer #3 (Remarks to the Author):

Thank you for revising the manuscript. I think that the changes that you made have strengthened the manuscript, and that your results will be of interest to many readers. I still find the discussion of the relationship between EM C/N ratio and ^{13}C or ^{15}N allocation to be confusing. It may be possible to address this with more clear and consistent language, additional details, and stronger syntax/paragraph structure. For example, it is difficult to follow the logic/progression of information in the section that begins on line 355. Additionally, regarding word choice-- the heading refers to "C/N ratio" but the following sentences refer to "C or N contents" plotted in Fig S5. Fig S5 shows the "C or N concentration" on the x axis and the "isotope enrichment" on the y axis. First--please add units to the y axis. Isotope values can be represented in different ways (atom %, atom % excess, delta, etc). It is important to specify this for the readers. Second--please add colors to indicate which fungal species is represented by each point in Fig S5. Third--ratio, content, concentration, etc are not necessarily interchangeable and can be confusing if not defined and used consistently. For example, to me, " ^{13}C enrichment decreased with increasing C content" (line 357) suggests that ^{13}C enrichment was negatively associated with total C content. But the data shown along the x axis of in Fig S5a are C concentration, not total C content. The relationship between ^{13}C enrichment and C concentration vs total C content may be different if EM tissues with high C concentrations have low biomass (and the relationship may vary when considered per EM root tip or per plant). Please be more precise in your discussion of C and N ratio, content, concentration, and enrichment throughout the manuscript. Overall, I think that your results indicate that EM C/N ratio and ^{13}C or ^{15}N allocation are associated, but this does not demonstrate that EM C/N ratio "drives" C or N allocation. Please adjust this claim here and throughout the text.

Reviewers' comments:

Reviewer #2

Overall, the revisions to this manuscript look very nice and the vast majority of my concerns have been alleviated. I reiterate that I believe this to be an interesting work that will be of interest to a wide readership.

Thank you for these positive comments on our study.

I have, however, two remaining issues:

1. The changes to Figure 4 look very positive and the coloration of the data points improves the clarity of the figure a great deal. However, I am unclear why the data presented in the original version of Fig. 4B seems to vary from that the in revised figure yet the associated equation has not changed. I will paste both versions below but perhaps this could be checked to ensure the figure is still correct.

Thank you for pointing this out. We have identified the error in an inadvertent mixing of X-axis data referring to ^{13}C and ^{15}N enrichments in the lateral roots, when producing the coloured figure. We have corrected the figure and conducted a thorough review of all other figures and tables to ensure their accuracy.

- 2. There is a bit of detail lacking in the stable isotope measurement and data analysis. For instance, the text mentions (line 243) that acetanilide was used as the standard which I presume to mean was used as both an isotope and compositional standard (is this correct?). If it was used as an isotope standard, were multiple acetanilides used that had different $\delta^{13}\text{C}$ and $\delta^{15}\text{N}$ values or was only one material used? Could the stable isotope values of these materials be provided and or referenced? What type of data calibration was used – just a single point correction of a multiple-point correction using a slope and intercept?**

We added this information under materials and methods: “An acetanilide standard (C = 71.09%, N = 10.36%, $\delta^{13}\text{C}$ = -29.6‰, $\delta^{15}\text{N}$ = -1.6‰) was measured after each 10th sample and used for point calibration. A second in-house standard (wild boar liver $\delta^{13}\text{C}$ = -17.3‰, $\delta^{15}\text{N}$ = 7.3) was used periodically to check for scaling. The standard materials were calibrated against international standards (International Atomic Energy Agency: IAEA-N1 and IAEA-N2 for ^{15}N and NBS18 and IAEA 600 for ^{13}C , <https://nucleus.iaea.org/sites/ReferenceMaterials/Pages/Stable-Isotopes.aspx>.”

- 3. Further, the data in Sup Fig S5 is also confusing as I don't believe the text defines what the authors intend by “Isotope enrichment”. Are they using this term interchangeably with APE as defined in lines 244-246 or are they using it in a different form? Can they apply a unit to this axis on the table (whether % or ‰ depending on how they define the term)?**

We acknowledge the confusion and apologize. Now, we provide a colour graph showing the individual mycorrhizal species (Supplement Fig. S5). We used two y-axis, which now clearly show the enrichments with units. The Fig. legends now reads as ““Relationship between ^{13}C enrichment (closed circles) and ^{15}N enrichment (open circles) with the C concentration (A) and the N concentration (B) in different EM fungal species.”. Under materials and methods, we clarified the definition of “enrichment” (= concentration of newly acquired ^{13}C per unit of biomass).

Supplement Fig. S5

Reviewer #3

Thank you for revising the manuscript. I think that the changes that you made have strengthened the manuscript, and that your results will be of interest to many readers.

Thank you for the encouraging comments.

- 1. I still find the discussion of the relationship between EM C/N ratio and ^{13}C or ^{15}N allocation to be confusing. It may be possible to address this with more clear and consistent language, additional details, and stronger syntax/paragraph structure. For example, it is difficult to follow the logic/progression of information in the section that begins on line 355. Additionally, regarding word choice-- the heading refers to "C/N ratio" but the following sentences refer to "C or N contents" plotted in Fig S5.**

We agree. We have now revised that section for better readability and comprehension. We also addressed the word choice inconsistency and the use of distinct terms referring to the concentration and amounts of stable isotopes (see below). We inserted an additional subheading to improve the structure of the section (L. 386).

- 2. Fig S5 shows the "C or N concentration" on the x axis and the "isotope enrichment" on the y axis. First--please add units to the y axis. Isotope values can be represented in different ways (atom %, atom % excess, delta, etc). It is important to specify this for the readers.**

We acknowledge the confusion and apologize. Now, we provide a colour graph showing the individual mycorrhizal species (Supplement Fig. S5). We used two y-axis, which now clearly show the enrichments with units. The Fig. legends now reads as ""Relationship between ^{13}C enrichment (closed circles) and ^{15}N enrichment (open circles) with the C concentration (A) and the N concentration (B) in different EM fungal species.". Under materials and methods, we clarified the definition of "enrichment" (= concentration of newly acquired ^{13}C per unit of biomass).

- 3. Second—please add colors to indicate which fungal species is represented by each point in Fig S5.**

Done, please, view the figure above

Third--ratio, content, concentration, etc are not necessarily interchangeable and can be confusing if not defined and used consistently. For example, to me, “¹³C enrichment decreased with increasing C content” (line 357) suggests that ¹³C enrichment was negatively associated with total C content. But the data shown along the x axis of in Fig S5a are C concentration, not total C content. The relationship between ¹³C enrichment and C concentration vs total C content may be different if EM tissues with high C concentrations have low biomass (and the relationship may vary when considered per EM root tip or per plant). Please be more precise in your discussion of C and N ratio, content, concentration, and enrichment throughout the manuscript.

This is indeed an important comment. To address this concern, we have now consistently used two distinct terms throughout the manuscript.

- 1) **Isotope amount** (defined at line 258, 263) for the absolute amount or quantity of a specific stable isotope (i.e. ¹³C and ¹⁵N) present in a given sample of plant or ectomycorrhiza. The amount is measured in mg ¹³C and µg ¹⁵N (per plant)
- 2) **Isotope enrichment** (defined at line 256) for the concentration of a specific stable isotope per unit of biomass. It is shown as mg ¹³C or µg ¹⁵N per gram (g) dry weight. In line with other isotope studies, we used the term “isotope enrichment” rather than “isotope concentration” as we clearly specified in the methods that all calculations are based on the APE. Thereby, we have prevented any misinterpretation because our data refer to the concentration of newly acquired isotope per unit of biomass and not to the total concentration of this isotope per unit of biomass.

Overall, I think that your results indicate that EM C/N ratio and ¹³C or ¹⁵N allocation are associated, but this does not demonstrate that EM C/N ratio “drives” C or N allocation. Please adjust this claim here and throughout the text.

We understand the concerns of the reviewer. We have changed subheading and text (L.34, L. 367, L397, l. 486):

“Ectomycorrhizal ¹³C and ¹⁵N allocation are correlated with C/N ratio”

“In conclusion, we show that the C/N ratios of EM fungal species were correlated with plant-derived C and N allocation to root tips.”

“(…) ectomycorrhizal C/N stoichiometry may play a role in influencing the allocation of host resources to symbiosis.”

We kept the phrasing, where we cite literature that proposes EM-driven nutrient cycles.